# Gain and loss of function variants in EZH1 disrupt neurogenesis and cause dominant and recessive neurodevelopmental disorders

Genetic variants in chromatin regulators are frequently found in neurodevelopmental disorders, but their effect in disease etiology is rarely determined. Here, we uncover and functionally define pathogenic variants in the chromatin modifier *EZH1* as the cause of dominant and recessive neurodevelopmental disorders in 19 individuals. *EZH1* encodes one of the two alternative histone H3 lysine 27 methyltransferases of the PRC2 complex. Unlike the other PRC2 subunits, which are involved in cancers and developmental syndromes, the implication of EZH1 in human development and disease is largely unknown. Using cellular and biochemical studies, we demonstrate that recessive variants impair *EZH1* expression causing loss of function effects, while dominant variants are missense mutations that affect evolutionarily conserved aminoacids, likely impacting EZH1 structure or function. Accordingly, we found increased methyltransferase activity leading to gain of function of two *EZH1* missense variants. Furthermore, we show that EZH1 is necessary and sufficient for differentiation of neural progenitor cells in the developing chick neural tube. Finally, using human pluripotent stem cell-derived neural cultures and forebrain organoids, we demonstrate that *EZH1* variants perturb cortical neuron differentiation. Overall, our work reveals a critical role of EZH1 in neurogenesis regulation and provides molecular diagnosis for previously undefined neurodevelopmental disorders.

Neurodevelopmental disorders (NDDs) are a group of conditions that arise from impaired development of the central nervous system and include developmental delay and intellectual disability among the most severe manifestations. To date, hundreds of genes, predominantly affected by de novo mutations, have been implicated in the etiology of NDDs[1–3]. Approximately half of these mutations are nonsense, frameshift or splice site variants that cause loss of function, while the remainder are missense mutations with often unpredictable molecular consequences that are rarely interrogated at a functional level[3,4]. Thus, despite recent advances in next generation sequencing and molecular diagnosis, many variants remain pathogenically undefined, contributing to the lack of treatment options and diagnostic challenges for patients and their families[5].

A group of NDD genes particularly susceptible to pharmacological intervention are chromatin regulators, especially those encoding subunits of Polycomb Repressive Complex 2 (PRC2)[6–10]. The core PRC2 complex is formed by EED, SUZ12, and one of the two catalytic subunits, EZH1 or EZH2, that sequentially mono-, di- and tri-methylate the lysine at position 27 of the histone H3 (H3K27me1, me2 and me3 respectively). Through the establishment and propagation of H3K27me3, PRC2 maintains transcriptional repression of genes that regulate cellular identities during development and tissue homeostasis in the adult[11]. The involvement of PRC2 subunits and H3K27me3 demethylases, KDM6A and KDM6B, in the etiology and prognosis of cancers, has motivated an upsurge of pharmacological inhibitors with therapeutic potentials[12,13]. In addition, dominant de novo pathogenic

✉e-mail: aquizun@chop.edu

variants in *EZH2*, *EED* and *SUZ12* are a common cause of Weaver (OMIM #277590), Cohen-Gibson (OMIM #617561) and Imagawa-Matsumoto Syndromes (OMIM #618786) which are characterized by overgrowth and intellectual disability[14–16], and de novo variants in *KDM6B* and *KDM6A* are implicated in a syndromic NDD and Kabuki Syndrome (OMIM #147920 and #300867) respectively[17–19]. Notably, the current availability of H3K27 methyltransferase and demethylase inhibitors renders these H3K27me3-associated developmental disorders susceptible for therapeutical targeting.

Despite evidence involving PRC2 in human development and diseases, the implication of EZH1 is less recognized. EZH1 has traditionally been considered the minor catalytic subunit of PRC2, in part because its histone lysine methyltransferase activity (HMT) is weaker than that of EZH2[20]. During development, EZH2 catalyzes most of the H3K27me3 that represses differentiation genes. Thus, upon deletion of *Ezh2*, *Ezh1* can only partially maintain H3K27me3 and transcriptional repression[21–23]. Consequently, *Ezh2* deletion is fatal for embryonic progression in mice[24]. In contrast, *Ezh1* deletion barely affects H3K27me3 maintenance at early developmental stages and is compatible with life, at least in mouse and zebrafish[25,26]. Nevertheless, a growing body of evidence suggests that EZH1 is important during differentiation and for postmitotic cells. For instance, EZH1 is required for myogenic gene activation during skeletal muscle differentiation, and for atrophy induced response of myotubes and neonatal heart regeneration in mice[27–30]. In addition, EZH1 represses multipotency of hematopoietic stem cells in favor of more lineage restricted embryonic progenitors[31] and a switch from EZH2 to EZH1 expression co-occurs with the onset of blood cell differentiation[32]. Indeed, the reduction of EZH2 and maintenance or increase of EZH1 expression during differentiation is common to most cell types, but its functional relevance remains poorly understood.

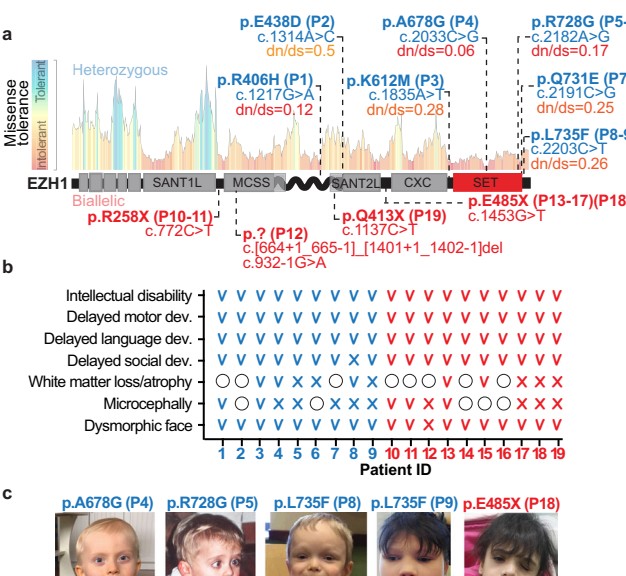

**Fig. 1 | Heterozygous and biallelic variants in *EZH1* are associated with previously undiagnosed neurodevelopmental disorders. a** Schematic representation of EZH1 protein and its domains with the missense tolerance landscape from MetaDome. The localizations of *EZH1* variants and the corresponding patient IDs are indicated. Heterozygous variants are labeled in blue with their missense tolerance score indicated. Biallelic variants are labeled in red and depicted below the EZH1 schematic representation. **b** Summary of major clinical features in patients with heterozygous variants (blue) and biallelic variants (red). V = present, X = absent, empty circle = unknown. **c** Photos showing dysmorphic facial features of patients carrying the indicated *EZH1* variants.

Similarly, little is known about the contribution of EZH1 to brain development and function. Unlike EZH2, which deletion in neural progenitors accelerates neurogenesis and promotes neuronal differentiation[33], EZH1 deletion does not cause overt cellular and molecular brain defects, at least in mice[34]. Nonetheless, evidence indicate that EZH1 is required for synaptic development of cultured mouse hippocampal neurons[35] and to protect neurons from degeneration upon EZH2 deletion[34].

Here, we show that *EZH1* is necessary for human brain development and function. Using next generation sequencing methods, we identified biallelic truncating and heterozygous missense *EZH1* variants in 19 individuals with varying degrees of developmental, language and motor delays, mild to severe intellectual disability and dysmorphic facial features. Notably, the lack of overgrowth as a hallmark distinguishes these patients from those with PRC2-EZH2 associated developmental syndromes. In biochemical studies and cellular models, we demonstrate that patients' *EZH1* variants cause recessive loss of function (LOF) or dominant gain of function (GOF) effects. Furthermore, through functional studies in the chick embryo neural tube development model and in genetically edited human pluripotent stem cells (hPSC) derived cortical neuron and forebrain organoids, we establish that *EZH1* LOF and GOF disrupt cortical neuron differentiation. Together, our work shows that a precise regulation of EZH1 activity is required to coordinate neurogenesis and uncovers *EZH1* LOF and GOF variants as the genetic basis of previously undefined NDDs with overlapping clinical features.

## Results

### Identification of *EZH1* variants in individuals with undefined neurodevelopmental delay

As part of our ongoing efforts to provide genetic diagnosis for patients with undefined neurodevelopmental disorders, we performed whole exome sequencing on a 15-month-old child showing cognitive, speech and motor development delay. The co-occurrence of the neurodevelopmental symptoms with an atypical facial dysmorphism suggested a genetic cause for the disease. Results pointed to a heterozygous missense variant in *EZH1* (NM_001991.5: c.2203 C > G; p.L735F) as the strongest candidate. Through collaborations facilitated by Genematcher[36], Deciphering Developmental Disorders project[37], and 100,000 Genomes Project[38] we expanded our cohort to 19 individuals from 14 unrelated families, all sharing a clinical phenotype of neurodevelopmental delay and carrying distinct variants in *EZH1* (Fig. 1, Supplementary Fig 1 and Supplementary Data 1), which supported the implication of *EZH1* in the disease pathogenesis.

Further supporting the pathogenicity of these variants, *EZH1* is a gene with low missense variant rate in the general population ($Z = 4.2$)[39] and constraints for loss of function variants (o/e = 0.26)[39] or gene duplication (pTriplo = 0.98)[40]. In addition, our patients' variants are absent in over 140,000 sequences of a sample population reported in GnomAD[39] and our internal databases, with exception of three variants (Supplementary Table 1). Among the variants identified, five are biallelic truncating variants present in 10 individuals (Fig. 1a). Two of the individuals are siblings of a consanguineous family carrying a homozygous nonsense variant (c.772C>T; p.R258X) inherited from their parents (Fig. 2a). Another homozygous nonsense variant (c.1453C>T; p.E485X) is present in four siblings and a relative of a large consanguineous family from Saudi Arabia. Interestingly, this family shares haplotype in chr17:40519338–41245021 with an additional affected child from an unrelated consanguineous Saudi Arabian family that carries the same homozygous variant, suggesting a common founder event in the region. Sanger sequencing in available family members confirmed segregation of *EZH1* variants according to a strict recessive mode of inheritance with full penetrance in these families (Fig. 2b, c). Furthermore, qPCR and Western blot (WB) analyses in three hPSC clones carrying p.E485X in homozygosity, showed that the

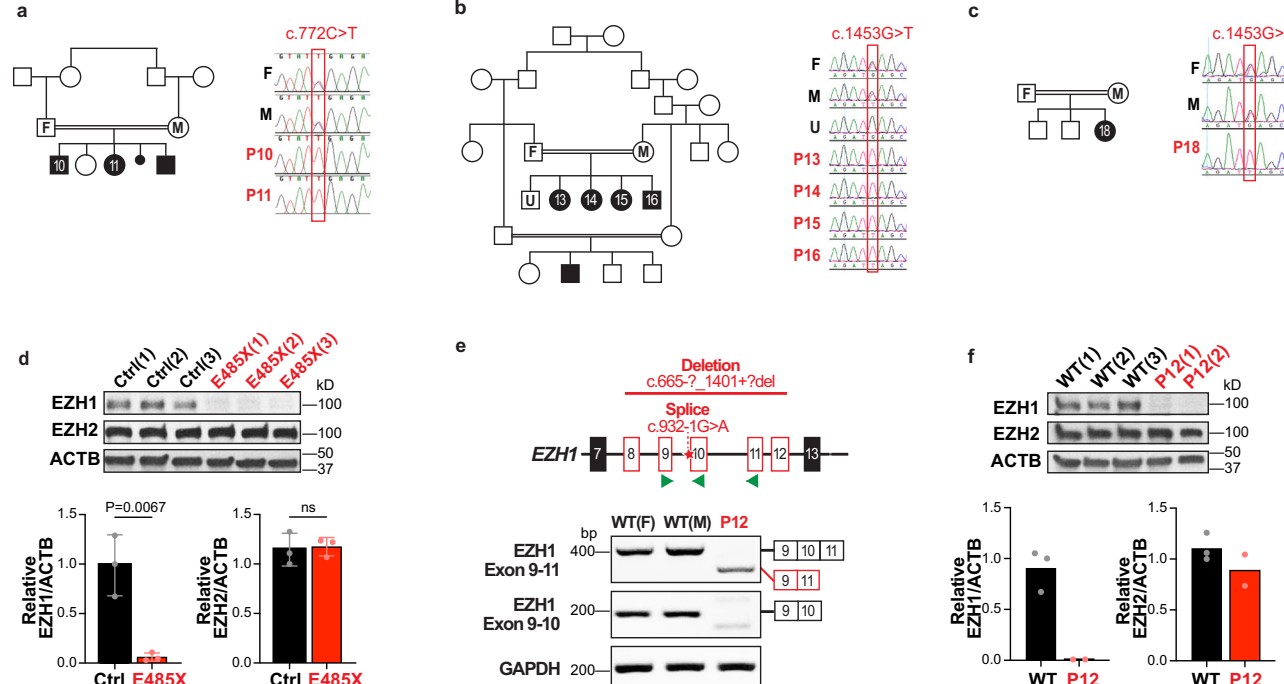

**Fig. 2 | Biallelic *EZH1* variants cause loss of function. a–c** Pedigree and Sanger sequencing showing segregation of variants with the diseases in consanguineous families with (**a**), 2 affected children (p10-11) harboring the homozygous nonsense *EZH1* c.772C>T; p.R258X variant (note that the youngest brother in the family was not included in the study) (**b**), 4 affected children (P13-16) harboring the homozygous nonsense *EZH1* c.1453G>T: p.E485X variant. and (**c**), a child (P18) from an unrelated family with shared haplotype and *EZH1* c.1453G>T: p.E485X variant. "F" indicates father, "M" mother and "U" unaffected family members. **d** Western blot analysis of EZH1, EZH2 and ACTB showing significant loss of EZH1 in hPSCs harboring the homozygous EZH1 p.E485X variant compared to isogenic controls. Graph shows mean ± SD of relative levels quantified by band densitometry in *n* = 3

independent clones. Two-sided unpaired *t* test. **e** Schematic representation of *EZH1* exon 7-13, indicating the location of deletion and splice variants in P12 and the primers used for RT-PCR with green arrowheads (top). RT-PCR results showing undetectable *EZH1* exon 10 containing transcripts in P12 lymphoblastoid cells compared to two unrelated wild type lymphoblastoid cells (WT(F) = females; WT(M) = male) and *GAPDH* as loading control (bottom). **f** Western blot analysis showing undetectable EZH1 levels and intact EZH2 in P12 cells compared to unrelated wild type (WT) cells. ACTB is shown as loading control. Graphs show mean relative EZH1/ACTB and EZH2/ACTB levels quantified by band densitometry in *n* = 3 WT cell lines and *n* = 2 P12 independent clonal cell lines. Statistical comparisons are not shown due to small sample size. Source data are provided as a Source Data file.

variant leads to loss of *EZH1* expression (Fig. 2d and Supplementary Fig. 2a). The ninth child carries the variant c.1137C>T; p.Q413X in homozygosity identified by solo exome sequencing (Supplementary Table 1). Finally, the only non-consanguineous case is a child that carries two variants in compound heterozygosity: a large deletion covering exon 8–12 of *EZH1* inherited from the mother (c.[664 + 1_665-1]_[1401 + 1_1402-1]del; p.?) and a de novo splice variant affecting the exon 10 splice acceptor site (c.932-1G>A; p.?) (Supplementary Fig. 2b, c). RNA extraction from patient's cells and analysis by RT-PCR confirmed exclusion of exon 10 (Fig. 2e), which predicts a frameshift that introduces a premature stop codon in *EZH1*, likely undergoing nonsense mediated decay. Accordingly, RT-qPCR of patient cells showed ~80% reduction of total *EZH1* transcripts (Supplementary Fig. 2d) and Western blot (WB) of cell lysates confirmed loss of full length EZH1 protein, which did not affect EZH2 expression levels (Fig. 2f). These data indicates that biallelic *EZH1* variants cause loss of function.

The remaining 9 patients carry heterozygous missense variants that affect evolutionarily conserved amino acids with low missense tolerance (Fig. 1a, Supplementary Fig. 3a). p.L735F and p.R728G are each present in two patients from unrelated families and the other five variants are unique to the affected individuals of five unrelated families. Peripheral blood DNA was obtained from parents of three families and absence of the pathogenic variants in the parents of two (for p.A678G and p.R728G) confirmed de novo germline origin. In the third family, the variant (p.K612M) was detected at low frequency in the father suggesting de novo somatic mosaicism with germline transmission to the affected child (Supplementary Fig. 2e).

To estimate the effect of the missense variants in EZH1 function we analyzed them through molecular modeling. Five variants fall within or near the catalytic SET domain, which conserves 94% similarity with EZH2-SET domain[11]. Although the 3D structure of EZH1 has been recently resolved[41], three of these variants fall within a gap in that structure. Thus, we used the 3D structure of EZH2-SET (PDB:5hyn[42]) as a template to model EZH1-SET domain and predict the effects of the new variants. With this strategy, we were unable to predict a pathogenic mechanism for p.Q731E given that the localization of the residue and the free energy calculations suggest a minor impact in the structure. However, results for p.K612M, p.A678G, p.R728G and p.L735F, show structural changes that may affect the kinetics of the methyltransferase reaction through different mechanisms (Supplementary Fig. 3b). For example, the aromatic ring of the phenylalanine that substitutes L735 residue can generate a stacking interaction with the adenine group of the methyl donor molecule (S-adenosyl methionine (SAM)) that could affect the turnover rate of SAM, and consequently, the efficiency of the H3K27me0 to me3 transition. Likewise, R728 is localized in the vicinity of the SAM binding site and close to the residues forming the binding pocket. We expect that its replacement by a glycine, a residue conferring more conformational flexibility, may induce a local structure destabilization and impact on SAM binding. On the other hand, the substitution of A678 residue for a glycine enlarges the H3K27 interaction pocket of EZH1 providing more flexibility to the structure and likely changing the stability of interactions with H3K27me0-me3. Visual analysis of the p.K612M variants shows that

the substitution to a methionine could disrupt the intra-monomeric interactions of the native lysine, thus locally destabilizing the structure of EZH1. Alternatively, the location of 612 residue in the surface of EZH1 suggests that the methionine could also interfere with the interaction of EZH1 with other proteins. Since K612 and A678 residues are present in the recently resolved EZH1 structure[41], we also used this structure to predict the effects of their substitutions. Results confirmed our previous predictions of p.K612M and p.A678G in EZH1 structure (Supplementary Fig. 3b).

The remaining two variants fall near the MCSS-SANT2L loop of EZH1. This region contributes to the nucleosome binding of PRC2:EZH1 and differs in sequence similarity, structure and function from the equivalent region in EZH2[41]. The recently resolved 3D structures of PRC2:EZH1 bound and unbound to the nucleosome [PDB:7KSR and 7KSO[41]) allowed us to model the p.E438D variant, but not p.R406H due to a gap in the 3D structure. Visual analysis of EZH1 with E438 or D438 shows a change in the orientation of the side chain of the aspartic acid compared to the glutamic acid, particularly in the nucleosome bound PRC2:EZH1 structure (Supplementary Fig. 3b). Furthermore, the free energy calculations suggest that the aspartic acid may destabilize the PRC2:EZH1 complex or its interaction with the nucleosome. Together, these data suggest that missense variants likely impact EZH1 catalytic activity through different mechanisms that affect the interactions with H3K27me0-me3, SAM or the nucleosome.

## Patients show neurodevelopmental delay with variable clinical presentations regardless of variant type and zygosity

The 19 patients in our cohort share a neurodevelopmental disorder manifested early in life as global motor, speech and cognitive delay leading to intellectual disability, usually non-progressive and co-occurring with dysmorphic facial features (Fig. 1b, c, Supplementary Fig. 1a and Supplementary Data 1). Eleven patients (P3-6, P8-9, P13, P15, P17-19) had magnetic resonance imaging (MRI) and all show mild or unremarkable findings (Supplementary Fig. 1b and Supplementary Data 1). Other clinical findings are variable between patients, regardless of the zygosity or type of mutation. For example, P19 is diagnosed with autism and P9 has a history of severe autism that was first suspected at 1–2 months of age due to a poor eye contact and tracking. Autism spectrum disorder (ASD) is being investigated in P7 as well, but there are no concerns for ASD in other patients. Aggressive behavior is reported in P5-6, who carry the same heterozygous variant (p.R728G), in P10-11 siblings harboring the homozygous p.R258X variant and in P19 with p.Q413X. In addition, P8 started showing aggressive and obsessive-compulsive behaviors at 7 years of age. P5 and P9 are the only patients with macrocephaly (>97%ile), while P1, P3, P10-11 and P13 have microcephaly. P1 and P3, who carry heterozygous missense variants, and P12 with biallelic LOF variants have short stature, but patients P5, P9 and P19 are taller than the average for their age. Together, these clinical presentations suggest that biallelic LOF and heterozygous missense EZH1 variants cause similar NDDs with otherwise variable clinical features that are clearly distinguishable from overgrowth with intellectual disability syndromes associated with PRC2-EZH2 associated variants.

## Missense EZH1 variants promote hypermethylation of H3K27

Clinical similarities between patients led us to predict that the effect of the heterozygous EZH1 missense variants could be similar to that of LOF variants, possibly through a dominant negative effect on EZH1 function. Thus, considering that most missense variants cluster within or near the catalytic SET domain of EZH1, we chose three of these variants (p.A678G, p.Q731E and p.L735F) to stably express in a human neural stem cell line (ReNCells) and to monitor their effects on EZH1 expression and H3K27me3 levels. After sorting transduced ReNCells for EGFP co-expression we analyzed protein and histone lysates by WB. Results showed similar EZH1 protein levels for cells transduced with

wild type (WT) EZH1 or one of the three variants (Fig. 3a and Supplementary Fig. 4a). However, unexpectedly, p.A678G and p.Q731E displayed increased overall H3K27me3 levels compared to EZH1 WT expressing ReNCells (Fig. 3b). To validate this result, we performed chromatin immunoprecipitation and sequencing (ChIPseq) using H3K27me3 antibodies, which showed a genome wide increase in the proportion of H3K27me3 levels within peaks of p.A678G expressing cells compared to EZH1 WT (Fig. 3c and Supplementary Fig. 4b).

We next analyzed if increased H3K27me3 levels were also detectable by p.A678G and p.Q731E in heterozygous conditions (like in patients). Therefore, we obtained P4 patient fibroblasts (with heterozygous EZH1 p.A678G variant) and generated hPSCs carrying p.A678G or p.Q731E in heterozygosity (EZH1$^{+/A678G}$ and EZH1$^{+/Q731E}$) (Supplementary Fig. 4c). WB analysis of P4 fibroblasts showed a statistically non-significant trend of increased H3K27me3 levels in comparison to control fibroblasts (Supplementary Fig. 4d). However, in hPSCs, H3K27me3 levels were comparable between mutants and their isogenic controls (Supplementary Fig. 4e). Given that EZH2 predominates over EZH1 in hPSCs[22], which may mask the effect of EZH1 on H3K27me3 levels, we next decided to analyze H3K27me3 under conditions where the effect of EZH2 is lower. For instance, it is well established that EZH2 levels decline with cellular differentiation[27–32,34,35]. For consistency with disease relevance, we chose neurons as the differentiation model where to test H3K27me3 levels. We first analyzed EZH1 and EZH2 levels during standard hPSC to neuron differentiation and confirmed that EZH2 expression declines by the second week in differentiation, while EZH1 levels remain constant (Supplementary Fig. 4f, g). We next differentiated EZH1$^{+/A678G}$ and EZH1$^{+/Q731E}$ to neurons for four weeks, and after validating that EZH1 and EZH2 levels were comparable to controls (Fig. 3d and Supplementary Fig. 4g), we measured H3K27me3 levels. WB analyses revealed a modest non-significant increase of H3K27me3 in EZH1$^{+/A678G}$ and a statistically significant ~50% increase of H3K27me3 levels in EZH1$^{+/Q731E}$ compared to control neurons (Fig. 3e).

To further diminish the contribution of EZH2 and dissect the effect of EZH1 variants in H3K27me3 molecularly, we performed in vitro histone methyltransferase (HMT) assays with reconstituted PRC2-EZH1 complexes and assembled nucleosomes harboring predefined methylation states at H3K27. We first expressed the PRC2 subunits EED, SUZ12, RBAP48, AEBP2, together with EZH1 WT or one of the three variants (p.A678G, p.Q731E or p.L735F) in a baculovirus expression system and intact complexes were successfully purified through tandem affinity purification for all (Fig. 3f, g, Coomassie staining). Versions of PRC2 carrying EZH1 WT or mutants were next incubated with tritiated methyl donor (SAM[$^3$H]) and unmodified nucleosomes (H3K27me0) (Fig. 3f) or nucleosomes harboring dimethylated H3K27 (H3K27me2) (Fig. 3g) as substrate. Samples were run in SDS-PAGE and stained with Coomassie to control for protein levels in each reaction and subsequently transferred to a PDVF membrane to assess H3K27 methylation levels by autoradiography. Results confirmed that mutant PRC2-EZH1 complexes have enhanced HMT activity relative to the PRC2-EZH1 WT complex (Fig. 3f, g). Interestingly, each variant showed slightly different effects and catalytic mechanisms on HMT activity. While EZH1 p.L735F showed only a modest variable increase methylating H3K27me0, EZH1 p.Q731E was more efficient than EZH1 wt methylating both H3K27me0 and H3K27me2 substrates (Fig. 3f, g). In contrast, EZH1 p.A678G was hyperactive on H3K27me2 but hypoactive on H3K27me0 (Fig. 3f, g). This finding implies that EZH1 p.A678G leads to increased H3K27me3 only on nucleosomes harboring H3K27me2, that may be deposited by an EZH1 WT copy or by EZH2 in vivo. In line with these results, many cancer-associated EZH2 mutations exhibit similar kinetic changes that result in their GOF activity. In addition we found that EZH1 p.A678G is equivalent to the cancer-associated EZH2-p.A677G variant, which modifies the substrate preference of EZH2 leading to a change in H3K27me0 to me3 kinetics that results in increased H3K27me3

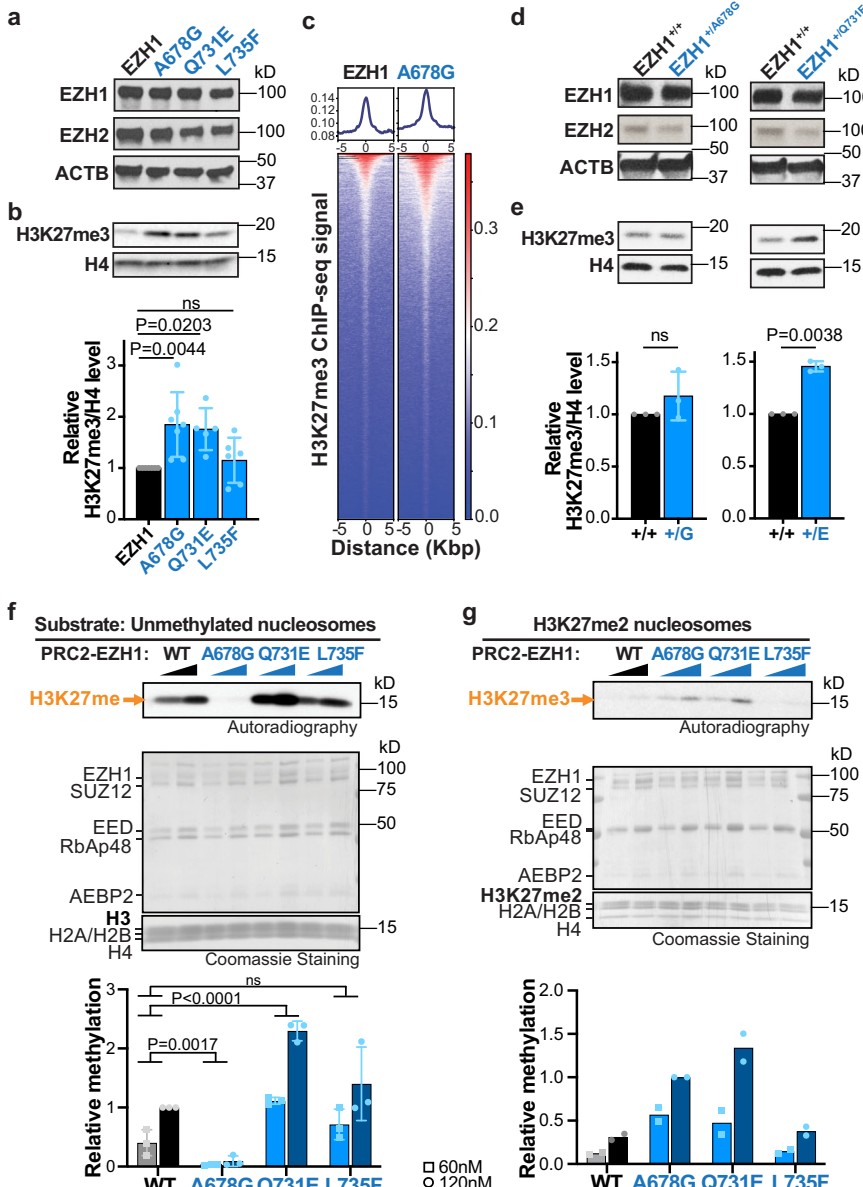

**Fig. 3 | Heterozygous missense variants cause EZH1 gain of function leading to hypermethylation of H3K27. a, b** Western blot analysis of EZH1, EZH2 (**a**) and H3K27me3 (**b**) in ReNcells transiently expressing either wild type or indicated *EZH1* missense variants. ACTB or H4 are shown as loading controls. Graph shows mean ± SD of H3K27me3/H4 levels quantified by band densitometry in *n* = 7 EZH1, *n* = 7 p.A678G, *n* = 5 p.Q731E and *n* = 6 p.L735F independent transductions. ns = non-significant. One-way ANOVA with Dunnett's post hoc analysis test for multiple comparisons. **c** Enrichment plots showing average signal of H3K27me3 in ChIPseq peaks (top) and heatmaps showing normalized H3K27me3 ChIPseq intensities (bottom) ±5 kb around the center of the peak in EZH1 or A678G expressing ReNcells. Plots represent data combined from 3 independent transductions. Two-sided paired t-test of signal in H3K27me3 peaks indicates statistically significant increase of H3K27me3 in A678G (*p*-value < 2.2e-16). **d, e** Western blot analysis of EZH1, EZH2 (**d**), and H3K27me3 (**e**) in 4-week old neurons derived from hPSCs

carrying EZH1 p.A678G or p.Q731E variants in heterozygosity (EZH1^{+/A678G} (+/G), EZH1^{+/Q731E}, (+/E)) or their isogenic controls (EZH1^{+/+}). ACTB or H4 are shown as loading controls. Graph shows mean ± SD of relative H3K27me3/H4 levels quantified by band densitometry in *n* = 3 independent differentiations. ns = non-significant. Two-sided paired *t* test. **f, g** Autoradiography and Coomassie stains of HMT assay reactions using two increasing concentrations of PRC2 complexes and unmethylated nucleosomes (**f**) or nucleosomes with dimethylated H3K27 (H3K27me2) as substrate (**g**). Graphs show mean ± SD of relative methylation levels quantified by band densitometry in *n* = 3 (**f**) or *n* = 2 (**g**) independent assays. ns = non-significant. Two-way ANOVA with Dunnett's post hoc analysis test for multiple comparisons for main variant effect (**f**). Statistical comparisons are not shown for graph (**g**) due to small sample size. Source data are provided as a Source Data file.

levels[43,44]. Thus, collectively, these data demonstrate that, at least, some missense *EZH1* mutations create *EZH1* GOF variants with increased methyltransferase activity. Combined with *EZH1* LOF variants found in the recessive NDD cohort, our observations suggest that precise EZH1 activity regulation is critical for neural development and homeostasis.

## EZH1 is necessary and sufficient for neuronal differentiation in the chick embryo neural tube

Having established the molecular consequences of *EZH1* variants, we sought to investigate how loss or gain of EZH1 impacts neural development. For this purpose, we first took advantage of the chick embryo neural tube, an in vivo model that has been used traditionally to

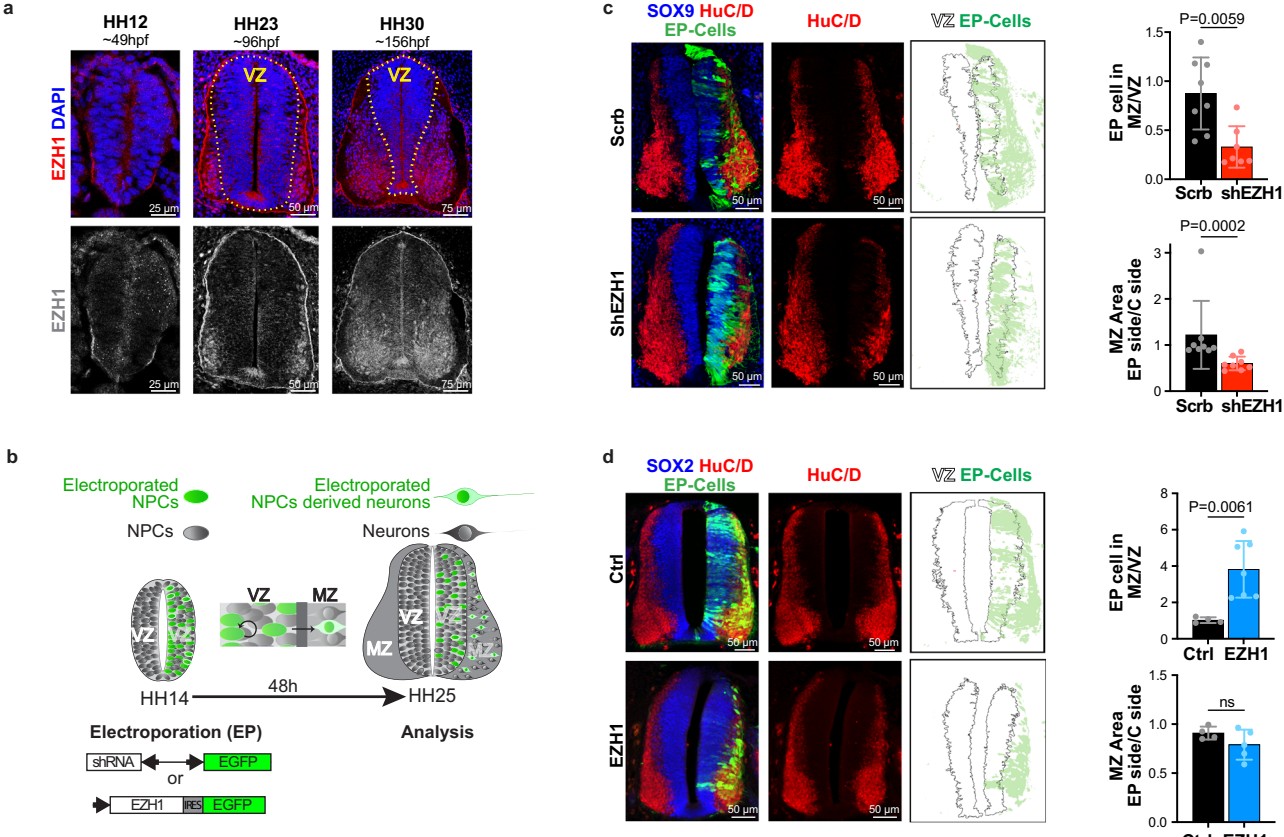

**Fig. 4 | Loss and gain of *EZH1* expression impairs neuronal differentiation during the chick embryo neural tube development. a** Images of immunostained transverse neural tube sections showing EZH1 enrichment in the mantle zone (MZ) neurons of HH23 and HH30 chick embryos. Dashed lines highlight ventricular zones (VZ) which are comprised by neural progenitor cells. Note that HH12 neural tubes are exclusively formed by neural progenitor cells. Images represent immunofluorescence results obtained from at least two embryos. **b** Diagram summarizing the *in ovo* neural tube electroporation procedure used to introduce DNA plasmids encoding indicated shRNA or cDNAs into neural progenitor cells of the HH14 chick embryos neural tubes. During 48 h after the electroporation, neural progenitor cells undergo proliferative divisions that produce more neural progenitors in the VZ and neurogenic divisions that generate neurons that delaminate to the MZ. Electroporated (EP) cells co-express EGFP and are shown in green. **c**, Images of transverse neural tube sections immunostained for SOX9 and HuC/D 48 h after the electroporation with DNA plasmids encoding EGFP and either a

scramble shRNA (Scrb) or a mix of two *EZH1* shRNAs (shEZH1). Right panels show EGFP signal over the VZ area perimeter highlighting predominant VZ localization of shEZH1 electroporated cells. Top graph shows the ratio of EP cells (EGFP⁺) located in the MZ vs in the VZ. Bottom graph shows HuC/D stained MZ area of the electroporated neural tube sides (EP side) normalized to the non-electroporated side (C side). Data represents the mean ± SD of $n = 8$ Scrb and $n = 7$ shEZH1 embryos. Two-sided Mann–Whitney U test. **d** Images of transverse neural tube sections immunostained for SOX2 and HuC/D 48 h after the electroporation with DNA plasmids encoding EGFP (Ctrl) or EZH1 and EGFP (EZH1). Right panels show EGFP signal over the VZ perimeter highlighting lack of EZH1 electroporated cells in the VZ. Top graph shows the ratio of EP cells (EGFP⁺) located in the MZ vs in the VZ. Bottom graph shows HuC/D stained MZ area of the EP side normalized to the C side. Data represents the mean ± SD of $n = 4$ Ctrl and $n = 7$ EZH1 embryos. ns = non-significant. Two-sided Mann–Whitney U test. Source data are provided as a Source Data file.

identify fundamental processes of vertebrate nervous system development and human disease[45–47]. At its formation (~33 h post fertilization, (hpf)), the chick embryo neural tube is exclusively comprised by neural stem cells that divide symmetrically to expand the progenitor pool within the ventricular zone (VZ). At ~60–70 hpf (~HH18 stage[48]), a switch from proliferative to neurogenic divisions of neural progenitor cells generates the first differentiating neurons that delaminate from the VZ and migrate laterally to form the mantle zone (MZ)[49]. This temporospatial segregation provides a convenient way to identify neural progenitor cells and differentiating neurons based on their position in the VZ and MZ of the developing neural tube respectively. Accordingly, by analyzing the expression pattern of EZH2 in transverse sections of the neural tube, we previously determined that EZH2 is predominantly expressed in neural progenitor cells of the VZ where it exerts important functions ensuring neural progenitor pool expansion, polarity and fate specification[50,51]. Here, we analyzed the expression pattern of EZH1 across the early neural tube development by immunofluorescence staining of transverse sections of HH12, HH23

and HH30 embryos. Results showed that EZH1 is undetectable in HH12 embryo neural tubes, which are exclusively comprised by neural progenitor cells. Similarly, EZH1 was barely detected in the VZ of HH23 and HH30 embryos. However, at these neurogenic stages, EZH1 expression was marked in the newly formed MZ of HH23 neural tubes and expanding MZ of HH30 neural tubes (Fig. 4a).

To determine if the predominance of EZH1 in the MZ is functionally relevant for differentiating neurons, we interfered with *EZH1* expression and monitored its effect in neurogenesis. For this purpose, we electroporated plasmids co-encoding EGFP and either a control shRNA (shScrb) or *EZH1* targeting shRNAs (shEZH1) into neural progenitor cells of HH14 neural tubes (before the onset of neurogenesis) and analyzed the effect in neuronal differentiation within a period of 48 h (Fig. 4b). As expected, the electroporation of two independent shEZH1s, successfully reduced *EZH1* expression (Supplementary Fig. 5a), likely by interfering with the upregulation of EZH1 in the electroporated (EGFP⁺) neural progenitors cells undergoing differentiation. Moreover, we noticed that the MZ area was smaller in

shEZH1 electroporated sides compared to shScrb electroporated or non-electroporated sides of the neural tubes. To further validate this observation, we labeled the MZ with anti-HuC/D differentiating neuron marker and confirmed a 50% reduction of the MZ (Fig. 4c) in neural tubes electroporated with shEZH1. This result suggests that interfering with *EZH1* expression during neurogenesis inhibits differentiation of neural progenitors to neurons. Accordingly, the proportion of shEZH1 electroporated (EGFP⁺) cells localized in the MZ was half of the electroporated cells within the SOX9⁺ neural progenitor cell pool of the VZ, in strike contrast to the shScrb electroporated EGFP⁺ cells that were evenly distributed between the VZ and the MZ (Fig. 4c). These data, combined with lack of phenotypes in apoptotic markers and mitotic cell numbers (labeled with active Caspase 3 and pH3 respectively) (Supplementary Fig. 5b, c) indicate that EZH1 upregulation is necessary for neural progenitor cells to differentiate and migrate to the MZ in the chick embryo neural tube.

Next, we overexpressed EZH1 in neural progenitor cells of HH14 neural tubes and assessed the effect on neuronal differentiation compared to the electroporation of a control plasmid (both co-expressing EGFP). 48 h after the electroporation, the control electroporated cells (EGFP⁺) were evenly distributed between the SOX2⁺ neural progenitor cell pool in the VZ and HuC/D⁺ neurons in the MZ, as expected by the switch from proliferative to neurogenic divisions of neural progenitor cells that occurs at -HH18[49], shortly after the electroporation (-16 h after electroporation). In contrast, the overexpression of EZH1 resulted in a significant increase of the electroporated (EGFP⁺) cells localized within the MZ relative to the VZ (Fig. 4d). Notably, the EGFP⁺ cells localized in the MZ of EZH1 electroporated neural tubes were also HuC/D⁺, suggesting that EZH1 overexpression induces neural progenitor cell differentiation and migration to the MZ. This phenotype is often caused by a premature switch of neural progenitor cells from proliferative to neurogenic divisions, which may cause a reduction of the neural progenitor cell pool and consequently, less differentiated neurons over time[45,46,49]. To test this possibility and also discard a contribution of neural progenitor cell death to the observed phenotype we next quantified pH3⁺ mitotic cell number, HuC/D⁺ mantle zone area and cleaved caspase3+ apoptotic cell number (Fig. 4d and Supplementary 5d, e). While none of these quantifications were significantly different between EZH1 and control electroporated neural tubes, more detailed analysis, including extended time windows which are limited by the transient nature of the electroporation, would be required to determine the consequences and the exact mechanisms by which EZH1 affects neuronal differentiation.

## *EZH1* is expressed in developing and adult cerebral cortex

Genes associated with neurodevelopmental disorders are expressed in the human cerebral cortex, particularly during the critical developmental periods of neurogenesis and synaptogenesis[10,52]. To ascertain if *EZH1* expression fits with this expression pattern, we examined developing and adult human brain transcriptomic databases. RNA sequencing data from the Genotype-Tissue Expression (GTEx) portal showed that *EZH1* is similarly expressed across different regions of the adult human brain, including the cerebral cortex (Supplementary Fig. 6a). We next explored the BrainSpan gene expression database[53], which includes RNA sequencing data from the human cerebral cortex at several pre- and postnatal stages. Results revealed a homogeneous and constant expression of *EZH1* across all the pre and postnatal stages. In contrast, *EZH2* showed a sharp drop during the developmental window that overlaps with neurogenesis, remaining low thereafter (Fig. 5a). These data suggest a possible central role for EZH1 as the predominant H3K27 methyltransferase beginning at cortical neurogenesis stages, and therefore, a concomitant vulnerability of the developing cerebral cortex to EZH1 loss or gain of function.

## *EZH1* LOF and *EZH1* GOF variants alter cortical neuron differentiation

To test whether *EZH1* LOF and GOF variants affect cortical neurogenesis, we took advantage of hPSC derived neurodevelopmental models, which provide unique and versatile resources to study mechanisms of neurodevelopmental disorders[54–56]. To achieve this goal, we generated isogenic hPSCs carrying an *EZH1* LOF mutation (EZH1⁻/⁻) and one of the patients' GOF variants (EZH1⁺/A678G) using CRISPR/Cas9 genome engineering technology and subsequently validated editing, pluripotency and genome integrity (Supplementary Fig. 6b–e). WB results confirmed loss of EZH1 in EZH1⁻/⁻ and intact EZH1 levels in EZH1⁺/A678G (Supplementary Fig. 6f).

To investigate the effects of EZH1⁻/⁻ and EZH1⁺/A678G in cortical neuron differentiation, we first generated cortical neural progenitor cells (NPC) from hPSCs using standard protocols[57]. As expected, all NPC cultures expressed PAX6 and NESTIN (Supplementary Fig. 6e). We also noted that EZH1⁻/⁻ NPCs continued expanding even after neuronal differentiation induction, while EZH1⁺/⁺ and EZH1⁺/A678G NPCs gradually stopped proliferating and acquired morphological features of differentiating neurons. To validate this observation, we first assessed proliferation rates by EdU incorporation and labeling at 0, 2 and 5 days after induction to neuronal differentiation. Consistent with our observation, the rate of EdU⁺ cells gradually decreased upon differentiation of EZH1⁺/⁺ and EZH1⁺/A678G NPC cultures. However, EZH1⁻/⁻ cultures showed similar proliferation rates across the 5 days in differentiation media (Fig. 5b), suggesting failure of EZH1⁻/⁻ NPCs to exit cell cycle and differentiate. To rule out the possibility that this result was due to inter-culture cell type heterogeneity, we co-labeled cells with the NPC specific marker SOX2, the proliferation marker Ki67 and the neuronal cell marker HuC/D at day 0 and 5 of differentiation and analyzed the percentages of proliferating NPCs (Ki67⁺ SOX2⁺) and differentiating neuronal populations (HuC/D⁺) by flow cytometry. The analysis confirmed that, regardless of the genotype, most cells are Ki67⁺ SOX2⁺ proliferating NPCs and HuC/D⁻ at day 0 (Fig. 5c, d, Supplementary Fig. 7a). At day 5 of differentiation, there was a marked decrease of the proliferating NPC population (Ki67⁺ SOX2⁺) in EZH1⁺/⁺ and EZH1⁺/A678G cultures, while in EZH1⁻/⁻ the NPC population was still large (Fig. 5c). Consistently, the increase of differentiating neuron population (HuC/D⁺) from day 0 to 5 was significantly lower in EZH1⁻/⁻ cells as compared to EZH1⁺/⁺ (Fig. 5d). Furthermore, when we assessed morphological features of differentiating neurons at day 2, EZH1⁻/⁻ cells exhibited short neurites resembling a less mature differentiation stage than EZH1⁺/⁺ and EZH1⁺/A678G differentiating neurons (Fig. 5e). Interestingly, this analysis also revealed the opposite effect in EZH1⁺/A678G cells, which showed longer neurites than EZH1⁺/⁺ cells (Fig. 5e) possibly representing a more mature stage in the differentiation process. Altogether this data indicates that EZH1 is necessary for the induction of neuronal differentiation from human NPCs.

Given the known function of PRC2 proteins coordinating the maintenance of gene repression programs during different stages of neural development[33,58,59], we anticipated that the effects on neuronal differentiation we observed with EZH1 LOF and GOF could be caused by dysregulation of genetic programs involved in diverse neuronal differentiation pathways. With the goal to identify these genes, we performed a transcriptomic analysis of 2-month-old neurons differentiated from EZH1⁺/⁺, EZH1⁻/⁻ and EZH1⁺/A678G hPSCs. At this differentiation stage, EZH2 expression is lower than in earlier differentiation stages and, consequently, EZH1 effects to the transcriptome could be exposed further. After performing principal component analysis (PCA) to confirm consistency between replicates and test similarities between samples (Supplementary Fig. 7b), we identified 852 down- and 546 up-regulated genes in EZH1⁻/⁻ and 1360 down- and 644 up-regulated genes in EZH1⁺/A678G neurons compared to controls (Fig. 5f, Supplementary Data 2–3). Gene ontology (GO) enrichment analysis on

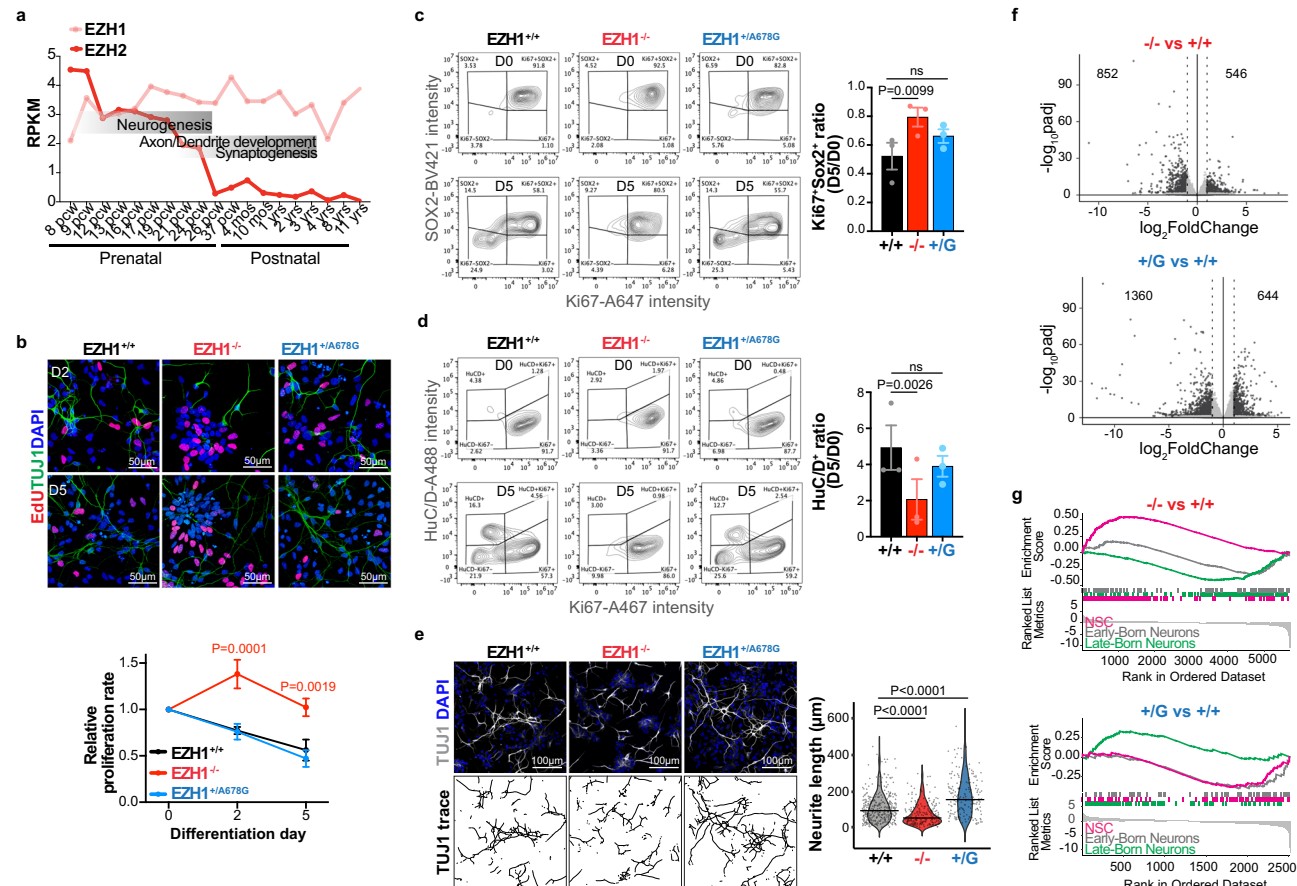

**Fig. 5 | EZH1 LOF and GOF variants alter hPSC-derived cortical neuron differentiation. a** *EZH1* and *EZH2* expression levels (RPKM) during pre- and postnatal cerebral cortex development. Data mined from mRNA-sequencing datasets in the Brainspan. **b** Images of EdU labeling and TUJ1 immunostainings at day 2 (D2) and 5 (D5) of NPC differentiation into neurons showing more EdU⁺ cells in EZH1⁻/⁻. Graph shows mean ± SEM of proliferation rates relative to D0 from *n* = 3 independent neuronal differentiations of hPSCs carrying EZH1⁺/⁺(+/+), EZH1⁻/⁻ (−/−) or EZH⁺/A678G (+/G). ns = non-significant. Two-way ANOVA with Dunnett's post hoc analysis test for multiple comparisons. **c** Representative flow cytometry contour plots of SOX2⁺ and Ki67⁺ cell populations at D0 and D5 of NPC differentiation into neurons. Graph shows mean ± SEM of D5 vs D0 ratio of SOX2⁺Ki67⁺ cell percentages from *n* = 3 independent differentiations. ns = non-significant. Two-sided paired *t* test with Holm−Sidak post hoc analysis test for multiple comparisons. **d** Representative flow cytometry contour plots of HuC/D⁺ and Ki67⁺ cell populations at D0 and D5 of NPC differentiation into neurons. Graph shows mean ± SEM of HuC/D⁺ cell percentage at D5 vs D0, for *n* = 3 independent differentiations. ns = non-significant. Two-sided paired *t* test with Holm−Sidak post hoc analysis test for multiple comparisons. Plots

illustrating the gating strategy used for flow cytometry analysis in (**c, d**) is shown in Supplementary Fig. 7a. **e** Images of TUJ1 immunostaining at day 2 of NPC differentiation into neurons show shorter neurites in EZH1⁻/⁻. Bottom panel shows traces of TUJ1 neurites. Violin plots show neurite length quantifications or *n* = 353 +/+, *n* = 228 −/− and *n* = 179 +/G cells recorded from 3 independent differentiations. One-way ANOVA with Dunnett's post hoc analysis test for multiple comparisons. **f** Volcano plots showing differentially expressed genes in EZH1⁻/⁻ (-/-) and EZH1⁺/A678G (+/G) compared to EZH1⁺/⁺ (+/+) 2-month-old neurons. Dotted vertical lines mark log2FoldChange = 1 and black dots represent genes with statistically significant changes (padj<0.05). Wald test with Benjamini−Hochberg method for multiple comparison. **g** Gene set enrichment analysis (GSEA) of neural stem cell, early-born and late-born cortical neuron gene sets, showing enrichment of neural stem cell gene set expression in EZH1⁻/⁻ (NES = 3.14, padj = 1E−10) and late-born neuron gene set in EZH1⁺/A678G (NES = 0.3, padj = 0.00026) compared to controls (EZH1⁺/⁺). Kolmogorov-Smirnov test with Holm method to adjust for multiple comparisons. Source data are provided as a Source Data file.

downregulated genes showed only terms unrelated with neuronal differentiation and highlighted few significantly enriched terms among the upregulated genes. Interestingly, genes upregulated in EZH1⁻/⁻ were significantly enriched for GO terms associated with mitosis while positive regulation of differentiation was among the top significantly enriched term in EZH1⁺/A678G neurons (Supplementary Fig. 7c).

Following this observation, we focused our enrichment analysis on genes specifically expressed along cortical neural development using Gene Set Enrichment Analysis (GSEA) and curated gene sets that included cortical neural stem cell markers, early-born neuron markers and late-born neuron markers (see methods section). Consistent with our previous observations, neural stem cell marker gene set was significantly enriched with EZH1⁻/⁻ upregulated genes, while differentiated neuron gene sets were downregulated (Fig. 5g). In contrast, neural stem cell gene set was enriched with genes downregulated in

EZH1⁺/A678G vs EZH1⁺/⁺, while differentiated neuron markers, and specifically the late-born neuron gene set, was enriched with upregulated genes (Fig. 5g). Interestingly, *SATB2* and *CUX1*, which are two genes expressed in late born cortical neurons, were among the genes upregulated in EZH1⁺/A678G vs EZH1⁺/⁺ (Supplementary Fig. 7d and Supplementary Data 3). This data supports that *EZH1* variation alters neuronal differentiation, with *EZH1* LOF enriching progenitor stages and *EZH1* GOF favoring features of differentiated neurons.

### *EZH1* LOF and GOF variants alter neurogenesis in forebrain organoids

We next sought to determine if defects in neuronal differentiation we observed with *EZH1* LOF and GOF variants were correlated with changes in cortical neurogenesis. Although monolayer cortical neuron cultures generate large amounts of grossly homogeneous cells ideal

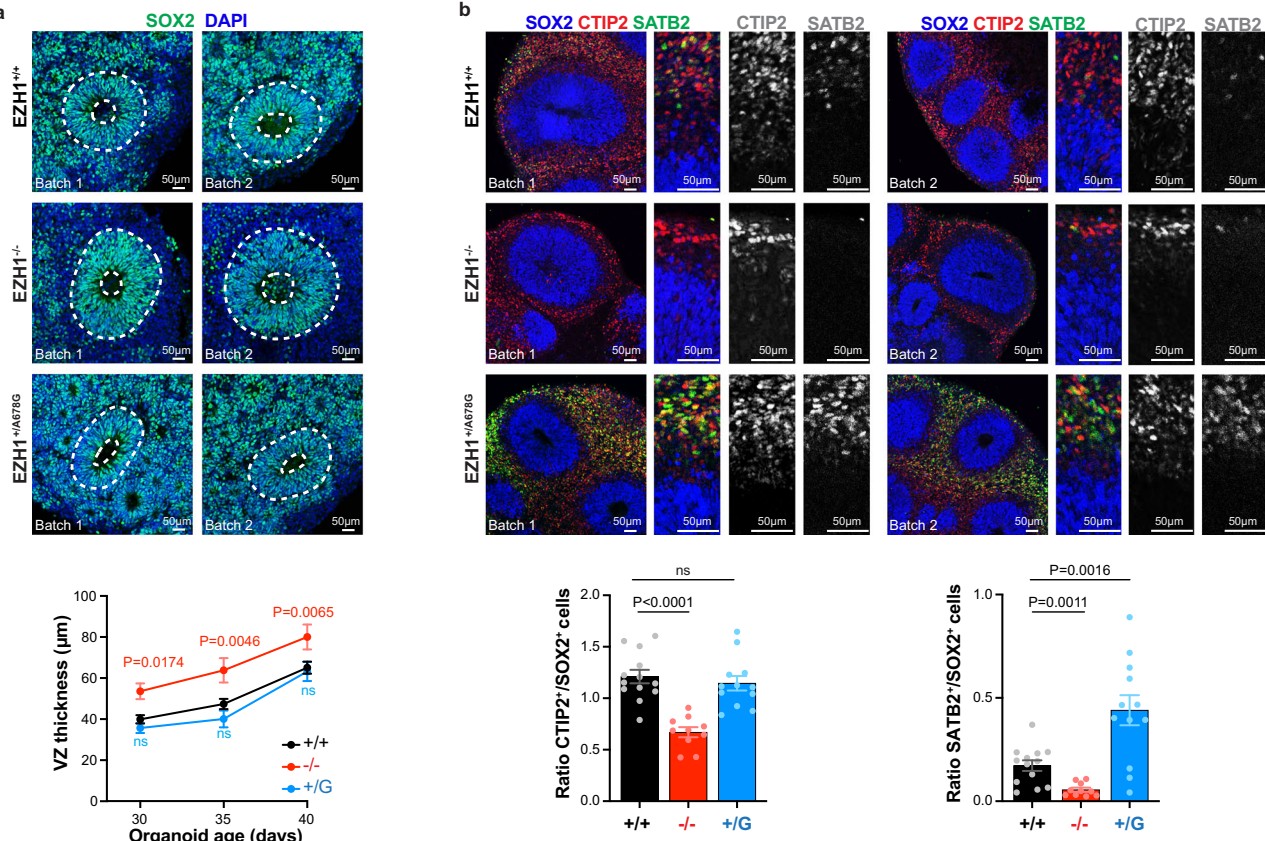

**Fig. 6 | Forebrain organoids derived from *EZH1* LOF and GOF hPSCs show defects in neurogenesis. a** Images of 35-day old forebrain cortical organoids immunostained for SOX2. Dashed lines highlight ventricular zones (VZ), which are thicker in EZH1$^{-/-}$ organoids. Graph shows mean ± SEM of VZ thickness in 30 ($n = 40$ +/+, $n = 25$ −/− and $n = 25$+/G), 35 ($n = 37$ +/+, $n = 24$ −/− and $n = 23$ +/G) and 40 ($n = 43$ +/+, $n = 26$ −/− and $n = 23$ +/G) day old organoids collected from two independent batches. ns = non-significant. Two-way ANOVA with Dunnett's post hoc analysis test for multiple comparisons. **b** Images of 60-day old forebrain cortical

organoids immunostained for SOX2 and CTIP2 (early-born cortical neuron marker) or SATB2 (late-born cortical neuron marker), showing less neurons in EZH1$^{-/-}$ organoids and more SATB2$^+$ neurons in EZH1$^{+/A678G}$ organoids. Graphs show mean ± SEM of the number of CTIP2$^+$ (left) and SATB2$^+$ cells (right) over SOX2$^+$ cells in $n = 13$ +/+, $n = 10$ −/− and $n = 12$ +/G organoids collected from two independent batches. ns = non-significant. Two-sided unpaired *t* test with Holm−Sidak post hoc analysis test for multiple comparisons. Source data are provided as a Source Data file.

for robust mechanistic studies, each analyzed timepoint captures the result of rather variable longitudinal differentiation process, which hinders our ability to determine if the phenotypes observed in each condition are caused by defects in neurogenesis or by other factors. In contrast, forebrain organoids are formed by self-organized neuroepithelial structures that hold neural progenitor cells in the ventricular zone (VZ) at the apical membrane and early and late born neurons progressively located near the basal membrane[60]. This spatiotemporal segregation of neurons and their progenitor cells makes it possible to assess neurogenesis rate at selected timepoints and compare them between conditions, by normalizing the number of differentiated neurons by the number of neural progenitors in each neuroepithelial structure. Thus, to test if *EZH1* LOF and GOF mutations cause defects in cortical neurogenesis, we generated EZH1$^{+/+}$, EZH1$^{-/-}$ and EZH1$^{+/A678G}$ forebrain organoid following a previously described protocol[60]. After a month in culture, all organoids showed neuroepithelial structures comprised by SOX2$^+$ NPC-containing VZs. While the thickness of the VZs was similar between EZH1$^{+/A678G}$ and EZH1$^{+/+}$ organoids, EZH1$^{-/-}$ exhibited thicker VZs (Fig. 6a). These data is consistent with defects in neuronal differentiation we observed in EZH1$^{-/-}$ NPCs monolayer cultures and suggests longer proliferation rates or deficits in neuronal differentiation.

We next analyzed 60-day old forebrain organoids. At this age, wild type forebrain organoids were comprised by SOX2$^+$ NPCs in the VZ, and CTIP2$^+$ early-born neurons around them. The first few SATB2$^+$ late-

born neurons were also detected in EZH1$^{+/+}$ organoids at this stage. Interestingly, the analysis of 60-day old EZH1$^{-/-}$ and EZH1$^{+/A678G}$ organoids revealed marked differences compared to wild type organoids. Specifically, EZH1$^{-/-}$ organoids showed less CTIP2$^+$ and TBR1$^+$ early-born neurons over the number of underlaying SOX2$^+$ NPCs (Fig. 6b and Supplementary Fig. 8a). In contrast, the relative amount of CITP2$^+$ neurons in EZH1$^{+/A678G}$ organoids was comparable to EZH1$^{+/+}$ organoids, but the amount of SATB2$^+$ late-born neurons was significantly higher than in EZH1$^{+/+}$ organoids (Fig. 6b). Although several neurobiological processes can lead to these phenotypes, lack of significant differences in cleaved Caspase3$^+$ cells between genotypes (Supplementary Fig. 8b) and lack of ectopically localized neurons or NPCs in neither EZH1$^{-/-}$ or EZH1$^{+/A678G}$ organoids (Fig. 6 and Supplementary Fig. 8a) discard defects in cell death or migration as the source of the neurogenesis defects and suggest that *EZH1* GOF and LOF variants affect the tempo of cortical projection neuron differentiation.

In summary, these results indicate that *EZH1* variants alter neurogenesis to specific cortical neuron populations which may ultimately result in defective development of neuronal networks causing the overlapping neurodevelopmental disorders associated with *EZH1* LOF and GOF variants.

## Discussion

In this study, we uncover recessive and dominant *EZH1* variants as the cause of overlapping neurodevelopmental disorders and demonstrate

an evolutionarily conserved function of EZH1 regulating neuronal differentiation. The recessive variants include three homozygous nonsense mutations predicted to impair *EZH1* expression in 9 children from 4 independent consanguineous families. We also identified a sporadic case with biallelic splice and deletion variants. The experimental confirmation of loss of *EZH1* expression in cells carrying either one of the homozygous nonsense mutations or the biallelic splice and deletion variants, indicates that recessives variants in *EZH1* cause LOF effects. These findings are consistent with a homozygous frameshift *EZH1* variant (c.237delT; p.L80Sfs*6) recently identified in two siblings through an independent genetic study[61]. Like patients in our cohort, the two siblings in this study were diagnosed with intellectual disability and showed dysmorphic faces. Combined, these data support that *EZH1* LOF variants are a likely cause of recessive NDDs.

The remaining 9 patients in our cohort all carry heterozygous missense *EZH1* variants that likely occurred de novo. To our knowledge, missense *EZH1* variants have never been associated with a human disease before, except for somatic EZH1 p.Q571R and p.Y642F mutations that are recurrent in minimally invasive thyroid and parathyroid tumors[62–64]. Although it is unclear how these *EZH1* mutations contribute to tumor formation or progression, the molecular characterization of EZH1 p.Q571R variant suggests a GOF effect[63] reminiscent of the missense mutations in our NDD cohort. GOF mutations in *EZH2* are also common in human cancers that motivated the development of EZH inhibitors as therapeutic agents[44]. Notably, one such mutation (i.e. EZH2 p.A677G) is equivalent to EZH1 p.A678G variant found in our patient P4[43,65,66]. Although none of our patients show malignancies, the molecular coincidence corroborates the GOF nature of the EZH1 p.A678G mutations in our NDD cohort. Nonetheless, it is possible that each missense variant affects EZH1 function in different ways. In line with this idea, our in vitro HMT assays revealed distinct effects on H3K27 methylation in the three EZH1 missense variants altering conserved SET domain residues that we tested (Fig. 3). Furthermore, two missense variants (p.R406H and p.E438D) fall far from the catalytic SET domain and near the EZH1 MCSS-SANT2L loop. Remarkably, the MCSS-SANT2L loop has recently been associated with an EZH1 specific ability to bind and condense chromatin in a methyltransferase activity independent manner[41]. This finding suggests a potential effect of variants p.R406H and p.E438D on EZH1 mediated chromatin condensation that warrants further investigation. Therefore, although our functional experiments show GOF effect for two missense *EZH1* variants, each missense variant will need to be carefully analyzed for effects on EZH1 activity.

Likewise, further work is required to better understand how *EZH1* LOF and GOF variants converge on overlapping neurodevelopmental phenotypes. Our work shows apparent opposing effects of EZH1 LOF and GOF in neural development. While EZH1 LOF leads to less differentiated neurons in the chick embryo neural tube and in hPSC derived forebrain cortical organoids, EZH1 GOF promotes neuronal differentiation. Interestingly, these opposing defects are consistent with an asynchronous neurogenesis that is emerging as a shared neurodevelopmental defect causing genetically heterogeneous NDDs[55]. Like in *KMT5B*, *CHD8* and *ARID1B* haploinsufficient brain organoids that were recently reported[55], EZH1 LOF impairs cortical projection neuron neurogenesis. Although the effect of EZH1 GOF in neurogenesis appears the opposite, premature neurogenesis of specific neurons could lead to asynchronous development of different neuronal classes ultimately converging with EZH1 LOF on abnormal development of neuronal circuits and consequent overlapping neurodevelopmental phenotypes. Regardless, our work demonstrates that cortical neurogenesis is vulnerable to EZH1 dosage and suggests that EZH1 LOF and GOF may alter the epigenetic transcriptional regulation of neurodevelopmental programs in a temporal and lineage specific manner.

Although other core PRC2 subunits have been genetically implicated in developmental syndromes[67], there are marked differences between these syndromes and the EZH1 associated NDDs identified in our study. Specifically, the hallmarks of patients with pathogenic variants in *EZH2*, *EED* and *SUZ12* are pre- and postnatal overgrowth with dysmorphic facial features and intellectual disability[67], while most patients in our EZH1 cohort have short or normotypic statures. Additionally, despite the overlap in the manifestation of intellectual disability, the neurobiological mechanisms causing neurodevelopmental defects are likely different. Pathogenic mechanisms that cause intellectual disability in PRC2 associated overgrowth syndromes remain unclear, however it is well established that PRC2-EZH2 maintains proliferative neural progenitor pools and regulates their fate transitions during development[33,50,59]. In contrast, our work shows that EZH1 is necessary for neuronal differentiation and neurogenesis of cortical projection neurons. Although additional work is needed to determine the exact neurobiological pathways involved in EZH1 mediated neurogenesis regulation, our data is consistent with a previously reported function of EZH1 in synaptic spine maturation and with the switch from EZH2 to EZH1 predominance that occurs as neural progenitors differentiate[35] (Fig. 5a). Together, these data support that EZH1 and EZH2 have non-redundant functions in neural development. However, future studies will need to elucidate whether the transition from proliferation of neural progenitors to neurogenesis is triggered by the decline of EZH2 expression or the predominance of EZH1. Furthermore, recent evidence indicates that PRC2 complexes can dimerize and colocalize in chromatin[41,68], thus, introducing a third possibility that includes the ratio between EZH1 and EZH2, or the relative levels of PRC2-EZH1 and PRC2-EZH2 homo and heterodimers, as determinants of cell fate transitions during neurogenesis.

NDDs caused by chromatin dysregulation are attractive candidates for treatment with epigenetic drugs. Recent years have witnessed an exponential increase in the development of epigenetic drugs, mostly purposed for the treatment of human cancers. Although these drugs are not recognized as treatment options for NDDs, the similarity of *EZH1* GOF variants with those found in *EZH2* associated cancers suggest a potential route for epigenetic recovery of *EZH1* GOF using EZH1/2 inhibitors. Furthermore, H3K27me3 demethylase inhibitors may have the potential to counterbalance *EZH1* or PRC2 LOF variants in NDDs. Treating NDDs presents several challenges, including the prenatal onset of the disease and the requirement for brain penetrant compounds. However, there are successful examples of postnatal reversal of neurodevelopmental phenotypes in animal models of Rett, Kabuki and Angelman syndrome[69–71], which hold promise for therapeutic targeting of PRC2 NDDs. Overall, our work uncovers a critical role of EZH1 in human neural development, provides a molecular diagnosis for patients with previously undefined NDDs and offers evidence to classify *EZH1* variants as recessive LOF or dominant missense with a likely GOF effect.

## Methods

### Ethical regulations and study participants

This study was approved by the Institutional Review Boards of the Children's Hospital of Philadelphia, Boston Children's Hospital, University College London, Guy's Hospital, Kennedy Krieger Institute, King Faisal Specialist Hospital & Research Center and University of Alabama in Birmingham. Ethical approval and informed consent were obtained for participation, phenotyping, sample collection and generation/derivation of patient and control cell lines including fibroblasts, lymphoblastoid cell lines, peripheral blood mononuclear cells and their reprograming. The authors also confirm that human research participants provided written informed consent for publication of the images in Fig. 1 and Supplementary Fig. 1. Since this study is focused on gene identification, all patients with confirmed diagnosis are included in the study, regardless of their gender and origin. Gender of each patient is indicated in Supplementary Data 1. Patients included in the study were identified based on genetic diagnostic through national

and international collaborations and belong to diverse origins, including Asia, Europe and USA.

## Identification of pathogenic EZH1 variants

Pathogenic variants in EZH1 were identified by exome or genome sequencing performed on blood genomic DNA extracted from the probands identified through diagnostic clinical practice or connections through GeneMatcher[30] and Deciphering Developmental Disorders Research Study[31] repositories. When available, blood DNA form parents and siblings was also sequenced. Data analysis was performed to assess quality of sequence reads and variants filtered with several pipelines[72–75] according to mode of inheritance, conservation, predicted pathogenicity and frequency in population. For P8 specifically exome sequencing was performed using the Agilent SureSelect Exome bait design V5 (Agilent, Santa Clara, CA, USA) and Illumina HiSeq2500 sequencer (Illumina Inc, San Diego, CA, USA) at the Center for Applied Genomics at the Children's Hospital of Philadelphia. The data was quality controlled and analyzed with BWA v0.7.12[76], Picard v1.97 (http://broadinstitute.github.io/picard), and GATK v2.6.5[77]. This resulted in a mean target coverage depth of 68.5x. ANNOVAR[78] and SnpEff[79] were used to functionally annotate the variants and collect minor allele frequency (MAF) data from 1,000 Genomes Project[80], ESP6500SI (http://evs.gs.washington.edu/EVS/), ExAC[81], gnomAD[39], and Kaviar[82]. Relatively common variants were excluded based on MAF threshold of 0.1% in either population dataset, and functional annotation such as synonymous, non-exonic, and non-splicing-altering were also included.

Variants were reported according to standardized nomenclature recommended by the Human Genome Variation Society[83] on EZH1 transcript GenBank: NM_001991.5. Genomic sequencing data in gnomAD[39] and our internal databases were used to determine the frequency of identified variants in the population and residue specific missense tolerance score retrieved from Metadome[84]. Conservation was determined by alignment of EZH1 and EZH2 protein sequences from different species using ClustalW2[85].

## Structural characterization of missense variants

For each variant we built a three-dimensional model using the package MODELLER[86]. The structure of EZH1 (PDB: 7KSO and 7KSR) was used as a template for the E438D variant. For variants R406H, R728G, Q731E, L735F, because 7KSO did not cover their locations, we followed a simple, two-step protocol. First, we created a model of the native EZH1, using the experimental structure of EZH2 (PDB: 5HYN) as a template. Then, we utilized the resulting model to build the structure of the desired variants. Variant A678G was also analyzed using this model, because it includes an H3K27me3 mimetic peptide which allows us to assess the effect of A678G on the catalytic pocket. In the case of L735F, to illustrate the potential stacking interaction between the aromatic side chain of the phenylalanine amino acid and the SAH molecule, we added local information from the structure of the SMYD3-SAM complex (PDB: 3MEK).

## RNA extraction, cDNA generation and PCR

Total RNA was extracted from patient and control LCLs using TRIzol Reagent (Invitrogen, # 15596026) and purified using RNeasy Plus Mini Kit (Qiagen). For splice site analysis, directed reverse transcription was performed using 0.5 µg of total RNA and 0.2 µM of reverse EZH1 or GAPDH primers (Supplementary Table 2) with SuperScript™ III First-Strand Synthesis System (Invitrogen, # 18080051). Standard PCR protocol was performed using 0.5 µM forward and reverse primers (Supplementary Table 2). Amplicons were ran in a 2% agarose gel and imaged using BioRad Gel Doc EZ Imager. For quantification of transcripts by RT-qPCRs, 0.5–2 µg of total RNA was retrotranscribed with oligodTs and SuperScript™ III First-Strand Synthesis System following standard protocol. Next, qPCR was performed in a 10 µL reaction volume containing 4.6 µL of cDNA sample, 0.5 µM of forward and revers primers (Supplementary Table 2) and 1X Power SYBR Green PCR Master Mix in a Biorad CFX384 Touch Real-Time PCR. Primers were designed using Benchling's Molecular Suite Primer3 program or NCBI Blast.

## In vitro histone methyltransferase assay

Standard HMT assays were performed in a total volume of 15 mL containing HMT buffer (50 mM Tris-HCl, pH 8.5, 5 mM MgCl$_2$, and 4 mM DTT) with 10 mM of $^3$H-labeled S-Adenosylmethionine (1:20 of $^3$H-labeled:unlabeled), 15 mM of H3K27me3, indicated amount of recombinant human PRC2-EZH1 complexes and 120 nM of either unmodified or H3K27me2 mononucleosomes. The reaction mixture was incubated for 60 min at 30 °C and stopped by the addition of 4 mL SDS buffer (0.2 M Tris-HCl, pH 6.8, 20% glycerol, 10% SDS, 10 mM β-mercaptoethanol, and 0.05% Bromophenol blue). After HMT reactions, samples were incubated for 5 min at 95 °C and separated on SDS-PAGE gels. The gels were then subjected to Coomassie blue staining for protein visualization or wet transfer of proteins to 0.45 mm PVDF membranes (Millipore). The radioactive signals were detected by exposure on autoradiography films (Denville Scientific).

## ReNcell VM culture

ReNcell VM human neural precursor cells (Sigma Aldrich, SCC008) were plated onto Matrigel (Corning, 354277) coated T75 cell culture flasks and cultured in ReNCells media containing 1:1 ratio of N2 and B27 media supplemented with 20 ng/mL Fibroblast Growth Factor (bFGF) (Thermo Fisher, #PHG0261) and 20 ng/mL Epidermal Growth Factor (bEGF) (Thermo Fisher, # PHG0261). N2 media contains DMEM/F12 (Gibco, #11330032), 1X N2 neural supplement (Gibco, #17502048), 5 µg/mL Insulin (MilliPore Sigma, #I9278), 1X GlutaMAX™ (Gibco, # 35050061), 100 µM MEM Non-Essential Amino Acids Solution (Gibco, #11140050), 100 µM β-Mercaptoethanol (Gibco, #21985023), 1 mM Sodium Piruvate (Gibco, #11360070) and 1% Penicillin-Streptomycin (Gibco, #15140122). B27 media contains Gibco Neurobasa medium (Gibco, #21103049) supplemented with 1X B-27 neural supplement (Gibco, #17504044), 1X GlutaMAX™ and and 1% Penicillin-Streptomycin. Cell culture media was changed every 2 days and cells passaged using Accutase (Gibco, #A1110501) when reached 90% confluency.

## Lentiviral production and transduction of ReNCells

Full-length cDNAs of human wildtype or variant EZH1 were cloned between the EF1A promoter and T2A-EGFP sequence in the pLV plasmid (VectorBuilder). For lentiviral production, HEK293-T (ATCC CRL-3216) cells were plated at $1 \times 10^6$ cell/well density in a well of a 6-well plate with 2 ml of media (1X DMEM (Gibco, #11995073) supplemented with 10% FBS (Tissue Culture Biologicals, #101), and 1% Penn/Strep (Gibco, #15140122)). The following day, cells were transfected using polyethylenimine (PEI) with 1.06 µg MDLg/RRE (gag/pol expression plasmid), 0.57 µg MD2.G (VSV-G envelope expressing plasmid), 0.4 µg pRSV-Rev (rev expression plasmid), and 1.6 µg of transfer vector (pLVs) per well. 48 h later 2 mL of media with lentiviruses were collected and filtered through 0.45 µm pore filters. 100–200 µL of filtered lentiviral media was added to ReNcells seeded the day before at 800,000 cells/well density in a well of a 6 well plate. ReNcells were incubated with lentiviruses for 48 h after which time media was replaced with fresh ReNcell media and cells cultured for 5 days before sorting with BD FACS Aria Fusion flow cytometer.

## Chick embryo neural tube electroporation

Eggs from white Leghorn chickens were eincubated at 38.5 °C and 70% humidity. Embryos were staged according to the method of Hamburguer and Hamilton (1951). In ovo electroporations were performed at stage HH14 (-54 h of incubation with 22 somites) with purified plasmid

DNA at 3 µg/ml in H2O and 50 ng/ml Fast Green. For EZH1 over-expression, *EZH1* cDNA was cloned between the *EF1A* promoter and *T2A-EGFP* in the pLV plasmid (VectorBuilder). *EZH1* shRNAs (shEZH1(1):GAAGTGTTCCAAGAAACGG, shEZH1(2): CAGTGTA-CAGTTGAGAGCA) were cloned into the pSHIN plasmid[87]. Electroporations were performed delivering five 50 msec pulses of 20–25 V. Embryos were collected 48 hrs post electroporation. Sex is not identified at these stages. According to the EU (where these experiments were performed) animal care guidelines no IACUC approval was necessary to perform the chick embryo experiments, since the embryos used in this study were all in early stages of embryonic development (between embryonic day 2 and 7).

### Generation of patient derived cell lines

P4 patient and unrelated control fibroblasts were established from skin biopsies cultured in MEM media (Gibco, 11095098) supplemented with 20% FBS (Tissue Culture Biologicals #101) and 1% Penicillin-Streptomycin (P/S) (Gibco, #15140122). P12 patient and unrelated control lymphoblastoid cell lines (LCLs) were established from peripheral blood lymphocytes and cultured and expanded in RPMI 1640 Medium (Gibco, #11875093) supplemented with 15% FBS (Tissue Culture Biologicals #101), 1X GlutaMAX (Gibco, # 35050061) and 1% P/S (Gibco, #15140122) in T75 flasks incubated at 37C in an upright position. P12 patient peripheral blood mononuclear cells (PBMCs) were isolated from fresh blood and reprogrammed with Sendai viruses expressing human Oct3/4, Sox2, Klf4 and c-Myc (Thermo Fisher Scientific, A16517).

### Pluripotent stem cell culturing

H9 human pluripotent stem cells (hPSCs) (WiCell WA09), KOLF2.1J (Jackson Laboratories JIPSC1000) and P12 iPSCs were cultured on Matrigel (Corning, #354277) coated plates in mTeSR1 media (STEM-CELL TECHNOLOGIES, #85850). 10 µM Rho-associated protein kinase (ROCK) inhibitor (Y-27632, Tocris, #1254) was only added into the culture media overnight when thawing the ESC/IPSCs. The medium was changed every day and cells were passaged when they reached 70% confluence, approximately every 5–7 days, using Versene (Gibco, #15040066) at a 1:15 or ReleSR (STEMCELL Technologies, #100-0484) at a 1:30 ratio.

### Generation of genetically modified hPSCs by CRISPR-CAS9

The 20 bp sgRNA sequences were cloned into the PX458 vector (Addgene, #48138). Single stranded DNA oligonucleotides (ssODNs) were synthesized by Integrated DNA Technologies (IDT). ROCK inhibitor was added to the culture media 24 h prior to electroporation. On the day of electroporation, Human Stem Cell Nucleofector Kit 1 (LONZA, #VPH-5012) was prepared based on the manufacturer's protocol. hPSCs were singularized with Accutase (Gibco, A1110501) and counted with Countess II Automated Cell Counter. To generate EZH1[+/A678G] clones, 3.5 µg PX458 plasmid (Addgene, #48138, from Feng Zhang) containing target sgRNA sequence and a mix of 1 µg ssODN containing the missense variant and two synonymous variants that disrupt the target sequence plus 1 µg of an ssODN containing only the two synonymous variants were transfect in $1.2 \times 10^6$ singularized hPSCs. To generate KO clones, 3.5 µg PX458 plasmid was transfect in $1.2 \times 10^6$ singularized hPSCs. To generate EZH1[E485X/E485X] clones, 3.5 µg of PX458 plasmid containing the target sgRNA sequence and 2 µg ssODN containing the nonsense mutation and a synonymous variant that disrupt the PAM sequence were transfected in in $1.2 \times 10^6$ singularized hPSCs. The Amaxa Human Stem Cell Nucleofector Kit I and B-016 nucleofector program was selected on Amaxa Nucleofector for electroporation. After electroporation, the cells were transferred to a Matrigel coated well with 10 µM ROCK inhibitor in the media. 24 h after electroporation, cells were fed with fresh mTeSR1 and 1% Penicillin-Streptomycin (P/S) (Gibco, #15140122). 48 h after electroporation, the

cells were singularized with Accutase, resuspended in mTeSR1 with 1% P/S, and filtered through 35 µm nylon mesh cell strainer cap tube (Falcon, #352235) and 1,000–2,000 GFP positive cells were sorted through the BD FACSAria™ Fusion Flow Cytometer, collected in mTeSR1 with CloneR supplement (STEMCELL Technologies, #05888), and seeded on a 10 cm dish coated with Matrigel. Single colonies were manually picked on day 8 to 10 and half of the colony was transferred to Matrigel coated 96-well plates and the other half processed for genomic DNA extraction followed by PCR and Sanger sequencing. Successfully edited clones were identified and two of them selected for experiments. Heterozygous clones were further validated by two rounds of serial single cell isolation, clone picking for sanger sequencing. EZH1[+/Q731E] hPSCs were generated by Synthego. The primers, sgRNAs, and ssODNs used for genome editing are listed in Supplementary Table 3.

### Cortical neuron differentiation

Cortical neurons were generated as previously described with few modifications[57]. Briefly, hPSC colonies were disassociated with Accutase, and seeded at ~50,000 cells/cm² density on Matrigel coated wells in mTeSR1 + 5 µM ROCK inhibitor. Next day media was replaced by neural induction media (50% DMEM/F12 and 50% Neurobasal, supplemented with N2 (Gibco, #17502048, 1:100), B27 (Gibco, A3582801, 1:50), GlutaMAX (Gibco, #35050061, 1:100), and 50 µM BME (Gibco, #21985023) 100 nM LDN193189 (STEMCELL Technologies, #72147) and 10 µM SB431542 (Stemgent, #04-0010-10)). 5 days later cells were singularized with Accutase and plated at a ~250,000 cells/cm² density in Matrigel coated wells, in neural induction media for additional 6 days. Next day cells were singularized again and plated at a ~250,000 cells/cm² on matrigel coated wells in neural progenitor expansion media (50% DMEM/F12 and 50% Neurobasal, supplemented with 1X N2, 1X B27, 1X GlutaMAX, 50 µM BME and 20 ng/ml bFGF). Media was replaced everyday. On Day 25, NPCs were singularized and plated on 100 µg/ml Poly-L-ornithine (PLO) (Sigma-Aldrich, P3655) and Matrigel coated wells in neuronal differentiation media (50% DMEM/F12 and 50% Neurobasal supplemented with 1X B27, 1X GlutaMAX, 50 µM BME, 10 ng/ml BDNF (Biothchne, #248-BD/CF), 10 ng/ml GDNF (Biothchne, #212-GD/CF), and 1 µg/ml laminin (Gibco, #23017015). Thereafter, ½ media was replaced with fresh media every other day.

### Cortical organoids differentiation

Forebrain organoids were generated as previously described[60]. On Day 0, iPSC colonies were detached by ReLeSR and aggregated to form EBs by Aggrewell (STEMCELL Technologies, #34811). The following day, EBs were resuspended and transferred to 6-well plate (Corning Costar) rotating at 110 rpm, containing DMEM/F12, 20% KnockOut Serum Replacement, 1X Non-essential Amino Acids, 1X GlutaMax, 1 µM LDN193189, 5 µM SB-431542 and 2 µg/mL heparin (STEMCELL Technologies). On Day 6, half of the medium was replaced with induction medium consisting of DMEM/F12, 1X N2 Supplement, 1X Non-essential Amino Acids, 1X GlutaMax, 1 µM CHIR99021 and 1 µM SB-431542. On Day 7, organoids were embedded in Matrigel and cultured stationarily in Low-attachment plate (Corning Costar) in the induction medium. On Day 14, organoids were mechanically dissociated from Matrigel by manual pipetting in a 10 mL pipette tip. Organoids were transferred to 6-well plate rotating at 110 rpm, containing differentiation medium consisting of DMEM:F12/Neurobasal, 1X N2 and 1X B27 Supplements, 1 × 2-Mercaptoenthanol, 1X Non-essential Amino Acids, and 2.5 mg/ml human Insulin. From Day 35 to Day 60, 1% Matrigel was supplemented in differentiation medium.

### Fluorescent activating cell sorting (FACS)

Cells were disassociated with Accutase™ (Gibco, #A1110501) and pelleted at 200 g for 5 min. Pellet was resuspended with 1 mL of 1X PBS with 10% FBS and filtered through a 35 µm nylon mesh cell strainer cap

into a 5 mL Falcon® Round-Bottom Tubes (Corning, #352235). Cells were loaded into the BD FACS Aria Fusion flow cytometer coupled with a BD FACSDiva Software (version 9.0.1). After excluding doublets or non-viable cells, GFP+ events were collected into a tube containing 0.5 ml of cell specific media.

## Flow cytometry analysis

Neural progenitor cells were plated at $1 \times 10^5$ cells/cm$^2$ density and let recover for two days. The third day NPCs were either harvested (for Day 0 analysis) or cultured in terminal differentiation media for five days (Day 5). For the immunostaining and flow analysis, cells were retrieved in Accutase, diluted with 1X DPBS, and centrifuged at 3000$g$, followed by resuspension and fixation in 1.6% paraformaldehyde for 30 min at 37 °C with agitation. Cells were subsequently washed once with 1X DPBS, resuspended into FACS Buffer (1X DPBS containing 0.5% BSA (Jackson ImmunoResearch) and 0.05% sodium azide) and stored at 4 °C until ready for analysis. Cells were then permeabilized with saponin buffer (diluted to 1X in water; Biolegend, # 421022) and incubated for 1 h in the following antibodies diluted in saponin buffer: anti-Ki67 (1:400; Cell Signaling, #9449), anti-HuC/D (1:200; Invitrogen, A-21271), and anti-SOX2 (1:300; Cell Signaling, #3579). Cells were subsequently washed with saponin buffer and incubated for 1 h with secondary antibodies goat anti-mouse IgG1-conjugated Alexa 647, goat anti-mouse IgG2b-conjugated Alexa 488, and goat anti-rabbit conjugated brilliant violet 421 (each at 1:500; Jackson ImmunoResearch 115-605-205, 115-545-207, 111-675-144, respectively). Cells were subsequently washed and resuspended into FACS buffer. Cells were analyzed using a CytoFLEX flow cytometer (Beckman Coulter) and the FlowJo software (version 10.8.2). The starting cell populations was selected using the area of the following coordinates (FSC-A, SSC-A): (146, 192k), (146k, 370k), (291k,642k), (551k, 932k), (906k, 1.1 M), (970k, 614k), (391k,112k), (192k,89k). These criteria selected 86 + −3% [SEM] of events of Day 0 samples, and 72 + −2% [SEM] of events of Day 5 samples, such that the area selected excluded debris and dead/dying cells. Positive and negative cell staining was defined as a direct comparison of the Day 0 vs Day 5 timepoints, such that the earlier timepoint (Day 0) determined the gating, knowing that the cells were proliferative (Ki67 + , SOX2 + ) and not yet terminally differentiated (HuCD-). The same gates, without modification, were applied to the Day 5 timepoint. Population groups and clustering were taken into consideration to be inclusive of the indicated group to provide appropriate gating as to not split a population.

## RNA-seq sample preparation and analysis

Total RNA was extracted using TRIzol reagent. RNA quality and concentration were determined with Bioanalyzer. Library preparation and sequencing was performed by Novogene on Illumina platform (NovaSeq 6000). Raw reads of FASTQ format were processed with fastp[88]. In this step, clean reads were obtained by removing reads containing adapter and >10% poly-N sequences, in addition to reads with low quality. At the same time, Q20, Q30 and GC content of the clean data were calculated. All the downstream analyses were based on the clean data with high quality. Reads were mapped to the reference genome GRCh37 (hg19) using STAR (v2.7.1a)[89]. The number of reads mapping to each gene were counted using featureCounts (subread-1.6.1[90]). Principal component analysis (PCA) of filtered and normalized counts were generated using ggplot2[91]. Differential gene expression was performed with DESeq2 (v1.28.1)[92] with count matrix. Volcano plots were generated with ggplot2[91]. GSEA were performed and plotted with ClusterProfiler (v.4.0.5)[93]. The input lists included 5705 DEGs from KO vs WT or 2517 DEGs from HET vs WT (padj<0.05, regardless of fold-change). Ranked cell-type specific gene lists composed of the top

500 significantly enriched genes in aRG, CFuPN, and CPN scRNA clusters were extracted from Uzquiano and Kedaigle et al[94] to generate the NSC, early-born neuron and late-born neuron gene sets (respectively) for GSEA analysis. Core enrichment lists were generated with GSEA. Heatmaps were generated with baseR. Gene ontology analyses were performed in Enrichr[95].

## Chromatin Immunoprecipitation, sequencing, and analysis

Cells were fixed with 1% formaldehyde for 10 min at room temperature. Fixation was quenched with glycine (0.125 M). Briefly, cross-linked chromatin was fragmented by sonication carried out using a Bioruptor to generate an average fragment size of 200–500 bp. Chromatin was purified by centrifugation for 30 min, at 20,000 $g$ and 4 °C. Purified chromatin was further diluted in immunoprecipitation (IP) buffer (0.01% SDS, 1.1% Triton X-100; 1.2 mM EDTA pH 8.0; 16.7 mM Tris–HCl pH 8.1; 167 mM NaCl) and incubated overnight with 1 μg of anti-H3K27me3 antibody (Cell Signaling (D5A7)). Protein A-bound beads were added and immunocomplexes were washed once with buffers TSE I (0.1% SDS; 1% Triton-X100; 2 mM EDTA pH 8.0; 20 mM Tris–HCl pH 8.1; 150 mM NaCl), TSE II (0.1% SDS; 1% Triton-X100; 2 mM EDTA pH 8.0; 20 mM Tris–HCl pH 8.1; 500 mM NaCl) and TSE III (0.25 M LiCl; 1% NP-40; 1% sodium deoxycholate; 1 mM EDTA pH 8.0; 10 mM Tris–HCl pH 8.1) and twice with TE buffer (10 mM Tris–HCl pH 8.1 and 1 mM EDTA). De-crosslinking was carried out for 4 h at 65 °C in elution buffer (1% SDS, 0.1 M NaHCO3). DNA fragments were purified using the QIAquick PCR Purification procedure to remove fragments >10 kb. DNA libraries were constructed with the TAKARA ThruPLEX® DNA-Seq Kit and the Illumina-compatible TAKARA DNA Single Index Kit. DNA libraries were quantified using a high sensitivity Chip on Bioanalyzer (Agilent) and sequenced at 40 million 150 bp pair-end reads per replicate in the NovaSeq 6000 (Illumina).

ChIP-seq reads were aligned to the GRCh37 hg19 reference genome using bowtie2 (v2.1.0) with parameters: -q --local --no-mixed --no-unal –dovetail[96]. Uniquely aligned reads (mapping quality score > = 20) and concordant alignments were kept for downstream analysis using command: samtools (v0.1.19) -q 20 -f 0 × 2. H3K27me3 peaks were called using MACS2 (v2.2.7.1) with parameters: --broad --keep-dup all -p 1e-5 --broad-cutoff 1e-5[97]. ChIP-seq signal was normalized to sequencing depth using deepTools (v3.5.0): bamCoverage --normalizeUsing CPM[98]. ChIP-seq signal around peaks was computed and visualized using deepTools computeMatrix and plotHeatmap functions, respectively. For the statistical test, H3K27me3 signal (RPKM) in peaks was calculated for both WT and A678G and paired t-test was performed to define the statistical significance of the H3K27me3 signal difference between WT and A678G ReNcells using ggplot.

## Western blotting

For histone-specific Western blotting, histones were enriched by lysing cells in an acid lysis buffer (10 nM HEPES, 1.5 mM MgCl$_2$, 10 mM KCl). For all other Western blots protein lysates were generated using RIPA Buffer (Cell Signaling, #9806). Protein lysates and histone extractions were separated by SDS-PAGE, transferred to a PVDF membrane, and detected by Western blotting (WB). Membranes were blocked with 5% BSA in 1X TBS with 0.01% Tween20 (TBST) for 1 h at room temperature and incubated with primary antibodies in 5% BSA in TBST overnight at 4 C (1:1,000 anti-EZH2 (Cell Signaling, D2C9), 1:1,000 anti-EZH1 (Proteintech, #20852-1AP); 1:1,000 anti-EZH1 (Reinberg lab[21]), 1:1,000 anti-H3K27me3 (EMD Millipore, 07-449), 1:5,000; anti-β-actin (GenScript, A00702), 1:10,000 anti-H4 (Abcam, ab10158)). The next day the membrane was washed three times with TBST, incubated for 1 h with the corresponding secondary anti-Rabbit HRP (Invitrogen, #31458) and anti-Mouse HRP (Invitrogen, #SA1-100), and washed three times with TBST. The membranes were developed using Pierce™ ECL Western Blotting Substrate kit (Thermo Scientific, #32106) and exposed to autoradiography films following development in AFP Mini-Med 90

X-Ray Fil Processor. ImageJ software was used for densitometry quantification of WB bands.

## Immunofluorescence staining

For chick embryo neural tube analyses, embryos were fixed for 2 h at 4 °C in 4% paraformaldehyde (PFA) and immunostaining was performed in 50 μm free floating sections cut with a vibratome. Sections were permeabilized with 1X PBS-0.1% Triton X-100, blocked with 5% BSA solution and immunostained with the following primary antibodies: 1:500 mouse anti-HuC/D (Millipore, A-21271) or 1:500 mouse anti-NeuN (Chemicon, MAB377) for MZ staining, 1:500 rabbit anti-SOX2 (Abcam, AB97959) or 1:500 rabbit anti-SOX9 (kindly provided by James Briscoe) to label the VZ, rabbit 1:500 anti-a-Caspase 3 (BD Pharmingen, C92-605) and 1:500 rabbit anti-Phospho-Histone3 (Millipore, #06-570) over night at 4 C. After washing three 5 min washes with 1X PBS-0.1% Triton X-100 sections were incubated with 1:1000 Anti-Rabbit Alexa Fluor 555 (Invitrogen, A21434), and/or anti-mouse Cy5 (Jackson ImmunoResearch, #115175146) secondary antibodies for 1 h at room temperature following by three 5 min washes. Sections were then stained with 1 μg/ul DAPI and mounted in Mowiol (Sigma-Aldrich, # 81381). Images were acquired with the Zeiss Lsm780 confocal system. The effects of *EZH1* shRNA electroporation were assessed by measuring the area occupied, respectively, by the VZ (formed by SOX2+ or SOX9+ progenitors) or the MZ (formed by HuC/D+ or NeuN neurons) in chick neural tube transversal sections. The analysis was adapted from[60]. Briefly, VZ or MZ channels were split and measured using the ImageJ software. After producing z-stack maximal projection images, a threshold was applied to define the areas in each side of the neural tube. VZ or MZ areas were measured in both the control and the electroporated side of the neural tubes by a particle analysis using a pixel^2 size ranging from 1000 to infinity. The data are presented as ratios of the area ± SD obtained by standardizing the values of the electroporated (EP) side with the corresponding values of the non-EP side of the neural tube. Similarly, the ratio of cells in the VZ and MZ was produced by manually counting the number of EGFP+ cells in the VZ and MZ in each neural tube.

hPSCs and derived neural cells were plated onto Matrigel coated glass coverslips (Electron Microscopy Sciences 72290-04). Cells were fixed with 4% PFA for 10 min at room temperature, permeabilized and blocked in blocking buffer (1X PBS-0.1% Triton X-100-5% goat serum (Sigma-Aldrich, G9023)) for 1 h at room temperature, and immunostained overnight at 4 °C with the following antibodies: 1:500 mouse anti-SSEA (Abcam, ab16287), 1:500 rabbit anti-OCT4 (Abcam ab19857), 1:300 rabbit anti-PAX6 (Biolegend, #901302), 1:2000 rabbit anti-Nestin (Millipore Sigma, MAB5326), 1:1000 rabbit anti-TUJ1 (Abcam, ab18207). Next day, cells were incubated for 1 h at room temperature with 1:500 anti-rabbit or anti-mouse Alexa Fluor 488, 555 or 633−conjugated secondary antibodies (Invitrogen, A11008, A21428, A21070, A11001, A21422, A21050) and 0.1 μg/ml Hoechst 33342 and mounted with Mowiol mounting medium (Sigma, #81381). For EdU (5-ethynyl-2′-deoxyuridine, Invitrogen, E10187) labeling, cells were treated with 10 μM EdU for 30 min. After fixation for 10 min at RT with in 4% PFA, EdU labeling was performed using Click-it plus EdU cell proliferation kit following the manufacturer's instructions (Invitrogen, C10638). Antibody co-staining was performed after EdU staining as indicated above. Images were acquired with Leica SP8 confocal microscope and analyzed in FIJI ImageJ. TUJ1 trace images were generated by FIJI ImageJ Process, Binary, skeletonize.

Whole organoids were fixed in 4% formaldehyde for 30 min at room temperature. Organoids were washed with PBS and then immersed in 30% sucrose solution overnight. Organoids were embedded in tissue freezing medium and sectioned with a cryostat (Leica) at 30 μm thickness. For immunostaining, cryosectioned slides were washed with 1X PBS. Tissues were permeabilized with 1X PBS-0.5% Triton-X in for 1 h and blocked with blocking medium consisting of 10% donkey serum in 1X PBS with 0.05% Triton-X (PBST) for 30 min. Primary antibodies 1:300 rabbit anti-SOX2 (R&D Systerms, AF2018), 1:300 rat anti-CTIP2 (Abcam, ab18465), 1:1000 mouse anti-SATB2 (Abcam, ab51502)) rabbit 1:1000 anti-TBR1 (Abcam, ab31940) and 1:400 rabbit anti-aCasp3 (Cell Signaling, #9661) diluted in blocking solution were applied to the sections overnight at 4 °C. After washing with PBST for 3 times, secondary antibodies (AlexaFluor 488, 546, or 647 -conjugated donkey antibodies (Invitrogen A21206, A31570, A78947)) diluted in blocking solution were applied to the sections for 1.5 h at room temperature. DAPI (Invitrogen) was added for 5 min (Invitrogen) at the end of incubation. Finally, sections were washed with PBST for 3 times before mounting. Images were captured by a confocal microscope (Zeiss LSM 800). Sample images were prepared in ImageJ (NIH). Total CTIP2+ or SATB2+ cells were counted in each image and normalized to SOX2+ cell numbers in the adjacent VZ. Analyses of VZ thickness was performed as previously described[60]. Briefly, to account for the variability, at least three organoids were sectioned for analysis. For each organoid, at least three sections distributed evenly from top to bottom of each organoid were sectioned and imaged. During confocal imaging, only rosettes at the edge of the organoids were captured to avoid the variability caused by the limited nutrient access in the center. Extremely small or big rosettes were avoided. Rosettes without clear lumen were also avoided as they may locate in very top/bottom region of the organoid. The distance from the apical to the basal lamina in a randomly selected position of each rosette in the section was measured for at least 3 sections per organoid.

## Statistics and reproducibility

The data were analyzed using GraphPad Prism software (Prism Version 9.1.0) or Microsoft Excell (Version 16.72). All experiments were performed at least twice and results reported as the mean ± SEMs or mean ± SDs with statistical tests described in figure legends when three or more biological replicates were analyzed. The investigators were not blinded to sample identity during experiments, but all measurements were taken objectively and no data were excluded. No statistical method was used to predetermine sample size.

## Reporting summary

Further information on research design is available in the Nature Portfolio Reporting Summary linked to this article.

## Data availability

All data supporting the findings of this study are available within the article and its supplementary information files. Source data are provided with this paper. The exome/genome sequencing will be made available upon request provided that privacy and consent criteria are preserved. The ChIPseq and RNAseq data generated in this study have been deposited in GEO under accession code GSE210465 and GSE227014. Other raw data will be made available within two weeks upon request to corresponding author. The human brain gene expression data used in this study are available in BrainSpan (https://www.brainspan.org/rnaseq/searches?exact_match=false&search_term=EZH1&search_type=gene and https://www.brainspan.org/rnaseq/searches?exact_match=false&search_term=EZH2&search_type=gene&page_num=0) and GTEX (https://gtexportal.org/home/gene/EZH1 and https://gtexportal.org/home/gene/EZH2). Variants p.R406H and p.A678G are accessible in ClinVar under the accession codes VCV000828189.1 and VCV000977755.2 respectively. The other variants characterized in this study have been submitted to ClinVar. Source data are provided with this paper.

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

## Acknowledgements

The authors thank the Center for Cellular and Molecular Therapeutics Stem cell core and Center for Applied Genomics sequencing cores at Children's Hospital of Philadelphia (CHOP) for their support with stem cell methods and sequencing technologies. We also thank Jasmine Akoto and Mara Davis, who helped with revisions during their PhD rotations in the Akizulab. The authors extend their appreciation to the invaluable contribution of Hessa Alsaif, other clinical coordinators, patients and their families to this study. This work was supported by the CHOP/UPENN IDDRC-New Program Development award, CHOP-Junior Faculty Pilot Program award, Margaret Q Landerbergen Foundation Award and NIH/NINDS R01NS119699, NIH/NICHD R21HD107592 (to N.A.), NIH/NINDS R35NS116843 (to H.S.), NIH/NINDS R35NS097370 and NIH/NIMH RF1MH123979 (to G.-l.M.)., BFU2015-69248-P and PGC2018-096082-B-I00 from the Spanish Ministry of Economy (to M.A.M.B.), Alabama Genomic Health Initiative F170303004 through University of Alabama at Birmingham (to A.C.H., M.T.), PID2019-111217RB-I00 Spanish Ministerio de Ciencia e Innovación (to X.D.l.C.), ID2019-110157RA-I00 (to M.S.), FPU Spanish Ministry of Education and Science predoctoral fellowship (to R.F.) and 2021 FISDU 00400 (to P.E-B.), NIHR Manchester Biomedical Research Centre NIHR203308 and Solve-RD which is supported by the European Union's Horizon 2020 research and innovation program grant agreement No 779257 (to S.B.). This research was made possible through access to the data and findings generated by the 100KGP and the DDD. The 100KGP is managed by Genomics England Limited (a wholly owned company of the Department of Health and Social Care). The 100KGP is funded by the National Institute for Health Research and NHS England. The Wellcome Trust, Cancer Research UK and the Medical Research Council have also funded research infrastructure. The 100KGP uses data provided by patients and collected by the National Health Service as part of their care and support. The DDD study presents independent research commissioned by the Health Innovation Challenge Fund [grant number HICF-1009-003], a parallel funding partnership between Wellcome and the Department of Health, and the Wellcome Sanger Institute [grant number WT098051]. The views expressed in this publication are those of the author(s) and not necessarily those of Wellcome or the Department of Health. The study has UK Research Ethics Committee approval (10/H0305/83, granted by the Cambridge South REC, and GEN/284/12 granted by the Republic of Ireland REC). The research team acknowledges the support of the National Institute for Health Research, through the Comprehensive Clinical Research Network.

## Author contributions

The first two authors, C.G.D. and Y.Z., contributed equally to this work. Study conceptualization and design: C.G.D., Y.Z., and N.A. Patient recruitment, clinical assessment and genetic diagnosis: R.M., D.L., S.E.S., P.M., M.H.H., H.Hakonarson, L.R., A.J., P.V., C.P., S.Mohammed, S.Maddirevula, H.A., E.A.F., S.E., V.T., F.R., S.Maqbool., V.S., S.H.I., G.d.R., H.Houlden., M.N.A., N.A.A., P.B., G.Z., A.C.E.H., M.L.T., A.C., C.L.S.-H. A.M.H., H.Z.E., M.H., C.R., D.B., S.B., G.E.R.C., F.S.A., P.B.A., and E.J.B. EZH1 in silico molecular modeling: X.dlC. and N.P. hPSC editing, culture maintenance, and differentiation: C.G.D., G.O., S.L., Y.Z., D.L.F., and N.A. Western blotting, RNA extraction, RT-PCR, RT-qPCR experimental design, execution, and data analysis: C.G.D. hPSC-derived neuronal culture immunofluorescence, image analysis and quantification: Y.Z. Histone methyltransferase assay: C.-H.L. and D.R. ChIP-seq experimental design and execution: C.G.D. ChIP-seq computational analysis: S.Z. and E.A.H. RNA-seq and computational analysis: C.G.D., Y.Z., and Z.S. Flow cytometry experiment conceptualization, execution and analysis: S.L., E.A.W., D.B., and N.A. Chick embryo experimental design, execution and data analysis: P.E.B., M.S., C.E., R.F., M.A.M.B., and N.A. Brain organoid culture, experiments and data analysis: Q.Y., H.S., G.-L.M., and N.A. Project coordination, supervision and administration: N.A. Manuscript preparation: C.G.D., Y.Z., and N.A. Manuscript edit and review: all authors.

## Competing interests

Houda Zghal Elloumi is an employee of GeneDX, LLC. Peter Bauer and Giovanni Zifarelli are employees of Centogene. The remaining authors declare no competing interests.

## Additional information

Carolina Gracia-Diaz [1,2], Yijing Zhou[1,2], Qian Yang [3,42], Reza Maroofian[4,42], Paula Espana-Bonilla [5,42], Chul-Hwan Lee[6], Shuo Zhang[7], Natàlia Padilla [8], Raquel Fueyo[5], Elisa A. Waxman [1], Sunyimeng Lei[1,2], Garrett Otrimski[1,2], Dong Li [9], Sarah E. Sheppard[9], Paul Mark [10], Margaret H. Harr[9], Hakon Hakonarson[9], Lance Rodan[11,12], Adam Jackson[13,14], Pradeep Vasudevan [15], Corrina Powel[15], Shehla Mohammed[16], Sateesh Maddirevula[17], Hamad Alzaidan[18], Eissa A. Faqeih[19], Stephanie Efthymiou [4], Valentina Turchetti[4], Fatima Rahman[20], Shazia Maqbool[20], Vincenzo Salpietro [4], Shahnaz H. Ibrahim [21], Gabriella di Rosa[22], Henry Houlden [4], Maha Nasser Alharbi[23], Nouriya Abbas Al-Sannaa[24], Peter Bauer[25], Giovanni Zifarelli[25], Conchi Estaras[26], Anna C. E. Hurst [27], Michelle L. Thompson[28], Anna Chassevent[29], Constance L. Smith-Hicks[29,30], Xavier de la Cruz[8,31], Alexander M. Holtz[12], Houda Zghal Ellouimi[32], MJ Hajianpour[33], Claudine Rieubland[34], Dominique Braun [34], Siddharth Banka[13,14], Genomic England Research Consortium*, Deborah L. French[1,2], Elizabeth A. Heller [7], Murielle Saade[5], Hongjun Song [3], Guo-li Ming [3], Fowzan S. Alkuraya[17,35], Pankaj B. Agrawal[12,36,37,38], Danny Reinberg[39], Elizabeth J. Bhoj[1,9], Marian A. Martínez-Balbás [5] & Naiara Akizu [1,2] ✉

[1]Raymond G. Perelman Center for Cellular and Molecular Therapeutics, The Children's Hospital of Philadelphia, Philadelphia, PA, USA. [2]Department of Pathology and Laboratory Medicine, University of Pennsylvania, Philadelphia, PA, USA. [3]Department of Neuroscience and Mahoney Institute for Neurosciences, University of Pennsylvania, Philadelphia, PA, USA. [4]Department of Neuromuscular Disorders, Queen Square Institute of Neurology, University College London, London, UK. [5]Department of Structural and Molecular Biology, Instituto de Biología Molecular de Barcelona (IBMB), Consejo Superior de Investigaciones Científicas (CSIC), Barcelona, Spain. [6]Department of Biomedical Sciences and Pharmacology, Seoul National University, College of Medicine, Seoul, South Korea. [7]Department of Systems Pharmacology and Translational Therapeutics, University of Pennsylvania, Philadelphia, PA, USA. [8]Research Unit in Clinical and Translational Bioinformatics, Vall d'Hebron Institute of Research (VHIR), Universitat Autonoma de Barcelona, Barcelona, Spain. [9]Center for Applied Genomics, The Children's Hospital of Philadelphia, Philadelphia, PA, USA. [10]Department of Pediatrics, Division of Medical Genetics, Helen DeVos Children's Hospital, Corewell Health, Grand Rapids, MI, USA. [11]Department of Neurology, Boston Children's Hospital, Boston, MA, USA. [12]Division of Genetics & Genomics, Boston Children's Hospital, Boston, MA, USA. [13]Division of Evolution, Infection and Genomics, School of Biological Sciences, Faculty of Biology, Medicine and Health, University of Manchester, Manchester, UK. [14]Manchester Centre for Genomic Medicine, St Mary's Hospital, Manchester University NHS Foundation Trust, Health Innovation Manchester, Manchester, UK. [15]Leicestershire Clinical Genetics Service, University Hospitals of Leicester NHS Trust, Leicester Royal Infirmary Leicester, UK. [16]Guy's Hospital, London, UK. [17]Department of Translational Genomics, Center for Genomic Medicine, King Faisal Specialist Hospital and Research Center, Riyadh, Saudi Arabia. [18]Department of Genetics, King Faisal Specialist Hospital and Research Center, Riyadh, Saudi Arabia. [19]Section of Medical Genetics, Children's Specialist Hospital, King Fahad Medical City, Riyadh, Saudi Arabia. [20]Developmental and Behavioral Pediatrics, University of Child Health Sciences & The Children's Hospital, Lahore, Pakistan. [21]Department of Pediatrics and Child Health, Aga Khan University Hospital, Karachi, Pakistan. [22]Child Neuropsychiatry Unit, Department of Pediatrics, University of Messina, Messina 98100, Italy. [23]Maternity and Children Hospital Buraidah, Qassim Health Cluster, Buraydah, Saudi Arabia. [24]John Hopkins Aramco Health Care, Pediatric Services, Dhahran, Saudi Arabia. [25]Centogene GmbH, Rostock, Germany. [26]Center for Translational Medicine, Department of Cardiovascular Sciences, Temple University, Philadelphia, PA, USA. [27]University of Alabama at Birmingham, Birmingham, AL, USA. [28]HudsonAlpha Institute for Biotechnology, Huntsville, AL, USA. [29]Department of Neurogenetics, Neurology and Developmental Medicine Kennedy Krieger Institute, Baltimore, MD, USA. [30]Department of Neurology, Johns Hopkins University School of Medicine, Baltimore, USA. [31]Institució Catalana de Recerca i Estudis Avançats (ICREA), Barcelona, Spain. [32]GeneDx, Gaithersburg, MD 20877, USA. [33]Division of Medical Genetics and Genomics, Department of Pediatrics, Albany Medical College, Albany, NY, USA. [34]Department of Human Genetics, Inselspital, Bern University Hospital, University of Bern, Bern, Switzerland. [35]Department of Anatomy and Cell Biology, College of Medicine, Alfaisal University, Riyadh, Saudi Arabia. [36]Division of Newborn Medicine, Boston Children's Hospital, Boston, MA, USA. [37]The Manton Center for Orphan Disease Research, Boston Children's Hospital, Boston, MA, USA. [38]Division of Neonatology, Department of Pediatrics, University of Miami School of Medicine and Holtz Children's Hospital, Jackson Heath System, Miami, FL, USA. [39]HHMI/NYU Langone School of Medicine, New York, NY, USA. [42]These authors contributed equally: Qian Yang, Reza Maroofian, Paula Espana-Bonilla. ✉e-mail: aquizun@chop.edu

## Genomic England Research Consortium

J. C. Ambrose[40], P. Arumugam[40], R. Bevers[40], M. Bleda[40], F. Boardman-Pretty[40,41], C. R. Boustred[40], H. Brittain[40], M. A. Brown[40], M. J. Caulfield[40,41], G. C. Chan[40], A. Giess[40], J. N. Griffin[40], A. Hamblin[40], S. Henderson[40,41], T. J. P. Hubbard[40], R. Jackson[40], L. J. Jones[40,41], D. Kasperaviciute[40], M. Kayikci[40], A. Kousathanas[40], L. Lahnstein[40], A. Lakey[40], S. E. A. Leigh[40], I. U. S. Leong[40], F. J. Lopez[40], F. Maleady-Crowe[40], M. McEntagart[40], F. Minneci[40], J. Mitchell[40], L. Moutsianas[40,41], M. Mueller[40,41], N. Murugaesu[40], A. C. Need[40,41], P. O'Donovan[40], C. A. Odhams[40], C. Patch[40,41], D. Perez-Gil[40], M. B. Pereira[40], J. Pullinger[40], T. Rahim[40], A. Rendon[40], T. Rogers[40], K. Savage[40], K. Sawant[40], R. H. Scott[40], A. Siddiq[40], A. Sieghart[40], S. C. Smith[40], A. Sosinsky[40,41], A. Stuckey[40], M. Tanguy[40], A. L. Taylor Tavares[40], E. R. A. Thomas[40,41], S. R. Thompson[40], A. Tucci[40,41], M. J. Welland[40], E. Williams[40], K. Witkowska[40,41], S. M. Wood[40,41] & M. Zarowiecki[40]

[40]Genomics England, London, UK. [41]William Harvey Research Institute, Queen Mary University of London, London EC1M 6BQ, UK.

