## [Peer Review File · Nature Communications]

REVIEWER COMMENTS

Reviewer #1 (Remarks to the Author):

Gracia-Diaz et al. has studied the implication of EZH1 overexpression and depletion in neurogenesis, likely leading to associated neurodevelopmental disorders. Using whole exome and genome sequencing and phenotypic characterization, they found distinct differences between patients, based on the type of genetic variant they harbor. Some of these results were further validated using chick model and forebrain organoids. Overall, the study is interesting and encompasses clinical as well as mechanistic approaches to understand the role of EZH1 in neurodevelopment. However, some aspects of this study need further clarification and analyses.

Major comments:

1. Fig4a and Lines 314-315: The claim the authors are making is not evident from the immunofluorescent image. Co-labeling with either progenitor or differentiated neuronal markers would be essential to substantiate the claim. Additionally, a higher magnification would be required to show presence or absence of colocalization.
2. Fig 5a: Reduction of EZH2 level can also be a trigger for neurogenesis. Also, the level to which this lowering happens is not significantly different from the level of EZH1 at the neurogenic time-points, is that correct? A clearer explanation or argument in the Discussion related to this would help.
3. Is considering length of neurite process in in vitro culture system enough to conclude about actual duration of neurogenesis and effect of EZH1 on it?
4. Lines 322-323: Smaller area is not always equivalent to less postmitotic neurons. The authors need to quantitate the neuronal density or its alterations as an effect of the mutations before coming to this conclusion.

Minor comments:

1. Supplementary Figure 3: EZH1+ cells are not clear. Adding a magnified image showing colocalization will be helpful.

Reviewer #2 (Remarks to the Author):

In this paper, Gracia-Diaz and colleagues report on the expression and role of Polycomb Repressive Complex 2 (PRC2) member EZH1 during neural development and its putative importance in human neurodevelopmental disorders. First, they report the identification of mutations in the EZH1 gene in 17 patients exhibiting apparently overlapping neuropsychiatric symptoms. In an attempt to provide further understanding of the molecular pathology caused by these mutations, the authors then use results from ChIP-Seq and Western Blotting experiments to suggest that some EZH1 genetic alterations found in patients are gain-of-function mutations that cause histone hypermethylation. Next, the authors embark on a series of experiments to phenotypically (histologically) analyze several cellular models of neurodevelopment (chick neural tube, differentiating human neurons in 2D culture and cerebral organoids) to suggest that neuronal differentiation is impaired after EZH1 expression levels or sequence are genetically manipulated.

Very little knowledge exists in the literature on the role of EZH1, particularly in the context of neurodevelopment, and evidence for its importance in clinical cases of neurological disorders is lacking. Therefore, this manuscript has the potential to fill important gaps of knowledge. However, the specific contributions of EZH1 in three fundamental processes that govern neural development (progenitor proliferation, neuronal differentiation and cellular migration) were not parsed out in this paper and therefore the conclusions drawn, especially in regard to the role of EZH1 during neuronal differentiation, are still immature and must be corroborated by further experimentation and controls.

A summary of the main points that require further experiments or clarification is provided below.

Major comments:

- 1) Why did the authors not include parents or unaffected siblings of matching sex as controls in the experiments in Fig. 3e-f? The use of unrelated controls is less than ideal, as a large inter-individual variability in gene expression is expected in these experiments.
- 2) The reportedly functional data (line 166 onwards) implicating the missense variants with loss-of-function of EZH1 is based on molecular modeling. In the absence of further confirmatory experiments in genetically modified cell lines (or patient-derived cells), these data are just suggestive and probably incomplete. This problem is compounded by the absence of molecular modeling data based on the resolved EZH1 structure itself (the EZH2 structure was used for evaluating most variants in the paper).

3) The MRI data in Fig. 1d is misleading because the text (lines 230-238) indicates that most patients do not display abnormal MRI findings. Therefore, these data should be moved to a supplementary figure, and the legend must clearly indicate that these findings are not the norm for most patients.

4) The section starting on line 228 attempts to describe the clinical presentation of neurodevelopmental abnormalities in all patients. This analysis makes it clear that the variability among patients is very large and no correlation was found with genotype (and probably could not, based on the small sample sizes for each mutation type). Therefore, I do not see this section as particularly revealing of the pathophysiology and disease mechanisms, which seem to be the paper's focus. I suggest that the section describing clinical characteristics be moved to the supplementary information or methods section. I also disagree with the last sentence in this section ("Additionally, differences in clinical features or severity between patients does (sic.) not correlate with the type or zygosity of EZH1 mutations") as no rigorous testing of correlation was performed (and probably could not, based on the small sample sizes).

(5) The experiments in Fig. 3a-b,d, in which EZH1 protein and histone methylation levels were assayed in a neural cell line over-expressing some missense EZH1 patient variants, are lacking quantification. This is an important piece of information. Proper replication and statistical analyses along with appropriate quantification (via for example band intensity analysis or preferably flow-cytometry) must be included.

(6) Likewise, the ChIP-Seq data in Fig. 3c is not replicated and statistical testing is lacking; a purely qualitative assessment is not ideal in this case, as these data are being presented as supporting evidence that the missense mutation increases EZH1's histone methylation activity.

(7) Although the experiments in Fig. 3e-f were nicely conducted, the quantification in Fig. 3f indicates the use of two replicates (n=2) and statistical power analysis is likely to indicate lower-than-expected sampling to achieve statistical significance in ANOVA followed by Dunn's post-hoc test.

(8) The design and interpretation of Fig. 4 results have many problems:

- the authors should improve their description of the developmental model used (chick neural tube), including a better description that HuC/D is used as a marker for the mantle zone (MZ) and SOX2/9 are used as markers of ventricular zone (VZ) neural progenitor cells.

- Suppl. Fig. 3 very clearly shows that EZH1 is expressed predominantly in the MZ. This makes the experiments in which the expression of EZH1 is manipulated in the chick neural tube's VZ very hard to interpret. The authors should comment on this matter and, if the results are kept in the paper, acknowledge this caveat.

- why was SOX9 used as a VZ marker for the shRNA experiments while SOX2 was used as a VZ marker for the EZH1 overexpression experiments?

- how can one differentiate between GFP+ electroporated VZ cells that then migrate to and differentiate in the MZ from the scenario in which GFP+ MZ cells are being electroporated in the first place?

- the data in Fig. 4c (fewer GFP+ cells in MZ and fewer HuC/D cells in MZ in shEZH1 neural tubes as compared to scrambled controls) is presented as evidence that "EZH1 regulates the balance between neural progenitor cells proliferation and differentiation during neural development" (line 329). This is certainly not the only explanation. EZH1 may impact VZ progenitor proliferation and/or VZ-to-MZ

migration and/or progenitor-to-neuron differentiation in the MZ. This model and experiments do not allow us to parse out between these three possibilities. Similarly, the EZH1 overexpression experiments are being presented as evidence that “EZH1 induces the *differentiation* of neural progenitor cells” (line 328), which is certainly unwarranted by the data.

- the statistical representation of data in Figs. 4c-d conflates technical replicates (histological sections from the same embryo) with biological replicates, which is inappropriate. Only the biological replicates should be presented and used for statistical purposes (all sections from the same individual should be aggregated into a mean per individual). Based on the SDs, it is likely that none of the mean difference comparisons shown will turn out to be statistically significant after this fix is applied (particularly in Fig. 4d).

- in the absence of molecular markers, it is not possible to define the exact location and cell types expressing EZH1 in the developing chick neural tube (Fig. 4a).

(9) In the experiments conducted in human neurons differentiated from hPSCs (Fig. 5), the CRISPR-derived mutant lines contain clinically relevant mutations only for the heterozygous variant. Why wasn't the loss-of-function(-/-) line produced with a clinically relevant truncating variant?

(10) The EdU+ cell counts in Fig. 5d are problematic, because the experiment is not controlled for the vagaries of the percentages of NPCs versus neurons generated in these types of 2D culture. Therefore, a measure of the diversity of cell types in these cultures must be included and factored into the calculations. Also, cytometry-based methods for determining the percentages of EdU+ cells are more appropriate.

(11) Although I consider the data on neurite length in Fig. 5d quite interesting and revealing of the potential cellular mechanisms derived from EZH1 loss-of-function, I disagree with the interpretation that these data reveal defects in neuronal *differentiation*. Neurite extension and growth is but one of many mechanisms during cellular differentiation and using that as a proxy for neuronal maturation is a dated approach. Rather, the authors should evaluate the expression of known neuronal maturation markers or analyze the transcriptomic landscape of these cells during differentiation induction in vitro (via RNA-Seq followed by RNA velocity analysis or pseudotime differentiation analysis). Moreover, it is hard to define whether differentiation is impaired if the cellular composition of differentiating cultures is not considered during the interpretation of these results, as pointed out above.

(12) Fig. 6a attempts at measuring the thickness of SOX2+ VZ-like neuroepithelial structures in cerebral organoids. However, these organoid derivation protocols lead to organoids with many such structures, which often differ in size and width (as can be seen in Fig. 6b). Moreover, the width of a particular VZ-like structure cannot be easily evaluated unless the experimenter analyses serial histological sections of each organoid. How did the authors select and quantify the VZ structure widths?

(13) I disagree with the conclusion from Fig. 6a that the larger width of VZ structures in organoids with EZH1 loss-of-function(-/-) reveal prolonged proliferation rates and deficits in neuronal differentiation (line 386). Either of these processes may be altered, but these experiments do not discriminate between both possibilities.

(14) The results in Fig. 6b show a deficit in CTIP2+ cortical-type neurons in EZH1-/- organoids and an apparent increase in SATB2+ neurons in EZH1 “gain-of-function” organoids. However, the SATB2 staining

seems odd. The percentages of SATB2+ neurons in the right-most panels are very high, which is unusual for 60-day organoids derived as per the protocol used. Moreover, immunostaining seems to label every cell between VZ-like neuroepithelia in “batch 2”. It was impossible to judge whether SATB2+ staining was overlapping with CTIP2+ staining in these cells, but it appears to be so. Therefore, proper controls, higher magnification panels, better quantification, and more replicates must be included to support the labeling quantifications in the right graphs.

Also, the protocol for organoid derivation (line 629) appears to conflict with the data reported in Fig. 6. If the differentiation factors (GDNF, BDNF, etc.) are added to the medium at day 70 in vitro, how do organoids at 60 days in vitro (Fig. 6) exhibit such high levels of differentiated cortical neurons?

More importantly, the conclusion drawn from these data -- that the different numbers of cortical neurons in genetically manipulated organoids indicate that EZH1 variants alter the timing of neurogenesis – is unwarranted. Many processes may lead to reduced/augmented numbers of cortical neurons, including abnormal NPC proliferation, migration, or cell death.

Minor comments:

- (1) Why did the authors perform RT-PCR and WB analysis on EZH1 for P12 only (lines 184-186 and in Fig. 2e-f)? Why were the other biallelic truncating variant individuals in the cohort not included?
- (2) The expression levels in control in Fig. 2e should have a normalized *MEAN* of 1, but each individual replicate measurement must be indicated, rather than normalized to 1.
- (3) Sequencing data and data supporting the genetic mosaicism in the father of one of the heterozygous affected individuals should be presented (related to lines 190-194) rather than just described in the text.
- (4) Some wording seems short of scientific precision, such as “diagnostic odyssey” and missense tolerancy (Fig. 1). Also, “catalyzes most of the H3K27me3” should be replaced with “catalyzes most of the methylation at H3K27” (there are similar instances throughout the rest of the manuscript).

Reviewer #3 (Remarks to the Author):

Gracia-Diaz and Zhou et al. reported that mutations in EZH1, which is part of PRC2 subunits, were identified from 17 individuals with neurodevelopmental delay. They found that biallelic recessive EZH1 variants led to LOF of its methylation activity, whereas heterozygous dominant variants lead to GOF. Interestingly, using a chick embryo neural tube model, they found that EZH1 down and upregulation both impaired neuron differentiation, but in opposite directions. They further supported this claim using hPSC derived isogenic 2D NPCs by showing EZH1 LOF variants led to delay neuron differentiation, whereas EZH1 GOF variant such as EZH1+/A678G promoted neuronal maturation. Lastly, they used a 3D forebrain organoid model to demonstrate that EZH1 GOF (EZH1+/A678G) and LOF variants affect neuron

differentiation by specifically impacting upper layer neuron and deep layer neuron production, respectively.

This paper reported novel findings on the role of EZH1 on neurodevelopmental delay. We have several minor comments for the authors to address.

Minor comments:

1. Typo in line 968, should be hPSC instead of hPCS.
2. For Fig. 2f, EZH2 level from P12(1) but not P12(2) is reduced compared to control. Can you explain this inconsistency?
3. Do you have references to support the claim in line 167?
4. Please add statistics for Fig.3e L735 vs WT comparison to support claim in line 296.
5. For Fig.5c and Fig.6, for the control line, did you use H9 unedited or H9+/+ synonymous lines as was indicated in supplemental Fig. 4e, which would then represent true isogenic with EZH1+/A678G, for 2D NPC and 3D forebrain organoid generation?
6. Based on supplemental table 1, the p.A678G variant was identified in the male patient P4. Why did you use H9, which is a female hESC line, to model this variant in Fig.5 and Fig.6?

RESPONSE TO REVIEWERS

We thank the reviewers for their encouraging and constructive feedback that have substantially improved our manuscript. Our revised manuscript includes new cell lines and experiments to further support that *EZH1* variants affect *EZH1* molecularly and lead to defects in neurogenesis that are likely associated with the neurodevelopmental delay of the patients with the NDDs we uncover. We have also revised the language of our manuscript to improve the description, conclusions and discussion of our experimental design and data as reviewers suggest. Below we include our point-by-point response to the reviewer comments:

Reviewer #1 (Remarks to the Author):

Gracia-Diaz et al. has studied the implication of *EZH1* overexpression and depletion in neurogenesis, likely leading to associated neurodevelopmental disorders. Using whole exome and genome sequencing and phenotypic characterization, they found distinct differences between patients, based on the type of genetic variant they harbor. Some of these results were further validated using chick model and forebrain organoids. Overall, the study is interesting and encompasses clinical as well as mechanistic approaches to understand the role of *EZH1* in neurodevelopment. However, some aspects of this study need further clarification and analyses.

We thank R1 for their insightful comments that we address below and in the revised manuscript.

Major comments:

1. Fig4a and Lines 314-315: The claim the authors are making is not evident from the immunofluorescent image. Co-labeling with either progenitor or differentiated neuronal markers would be essential to substantiate the claim. Additionally, a higher magnification would be required to show presence or absence of colocalization.

RESPONSE: We agree that colocalization with neural progenitor and neuronal markers would provide a more precise expression pattern of *EZH1*. However, in the developing neural tube we can easily distinguish neural progenitors and neurons based on their spatial localization: neural progenitors are localized in the ventricular zone (VZ) (in the center of the tube) and differentiating neurons in the mantle zone (MZ) (laterally adjacent to the ventricular zone) (See Fig R1). According to this, the more prominent expression of *EZH1* in the mantle zone (Fig 4a) indicates that *EZH1* expression is higher in differentiating neurons than in neural progenitors.

Nonetheless, we have tried to follow R1's suggestion to complement data in Fig 4a with neuronal marker colocalization studies, but unfortunately, we no longer have the *EZH1* antibody (home made in Reinberg lab) we used for Fig 4a and immunofluorescent stainings with 3 new *EZH1*

Figure R1: Chick embryo neural tube development model. a) Three developmental stages of the chick embryo neural tube showing Huc/D stained Mantle Zone (MZ) neurons in older stages (HH18 and HH24), while absent in the HH14 neural tubes that undergo electroporations. **b)** VZ and MZ of the HH24 neural tube are distinguished by the more intense DAPI staining of the ventricular zone (VZ) owing to a higher nuclear density than in the MZ. Red dashed lines in the bottom image mark the boundary between the VZ and MZ based on DAPI intensity, which correlates well with MZ labeling by Huc/D in the same HH24 neural tube section (figure 1a HH24).

antibodies we obtained have all failed (they either show background signal or no signal). However, to further support that EZH1 expression increases as neurogenesis progresses in the chick embryo neural tube, we now include in additional EZH1 immunofluorescence stainings we performed in the past with the original antibody. As shown in the revised *Fig 4a*, EZH1 is undetectable in HH12 (~49hpf) embryo neural tubes, which are only comprised by neural progenitors. HH23 (~96hpf) embryo neural tubes show EZH1 staining in their thin MZ; and in HH30 (~156hpf) neural tubes, EZH1 is also expressed in the MZ which is composed of more neurons and thus larger than the HH23 MZ. Please, note the similarity between EZH1 expression pattern in *Fig 4a* and HuC/D expression pattern in *Fig R1*. We hope that these additional data and the revised description of the chick embryo neural tube development model in the manuscript, further support that EZH1 is predominantly expressed in the MZ localized neurons.

2. Fig 5a: Reduction of EZH2 level can also be a trigger for neurogenesis. Also, the level to which this lowering happens is not significantly different from the level of EZH1 at the neurogenic time-points, is that correct? A clearer explanation or argument in the Discussion related to this would help.

RESPONSE: R1 is correct and in order to acknowledge the role of EZH2 in neurogenesis, the revised manuscript includes evidence reported in the literature, showing that deletion of EZH2 in neural progenitors accelerates neurogenesis (Pereira *et al.*, 2010; Akizu *et al.*, 2016).

As R1 suggests, we also extend the discussion to acknowledge that either the EZH2 decline or the EZH1 predominance could be a trigger of neurogenesis. Furthermore, we do not discard that a change in EZH1/EZH2 ratio may contribute to this transition (*discussed in lines 580-586*). These are interesting questions under research in our lab, however, with our current data, we can not determine if EZH2 levels are significantly lower than EZH1 at neurogenic timepoints, at least at protein level. At RNA level, based on RPKM levels reported in a single sample of Brainspan dorsal prefrontal cortex RNAseq (*Fig 5a*) the expression of EZH1 (RPKM=3.9) is higher than the expression of EZH2 (RPKMs=2.9) starting at postconception week 17 (around the onset of neurogenesis). However, the lack of additional samples precludes us to calculate the statistical significance of this difference. Thus, without further evidence, we prefer to avoid statements about relative abundance between EZH1 and EZH2 during neurogenesis and rather focus on the decline of EZH2 that begins with neurogenesis onset, while EZH1 expression is maintained constant.

3. Is considering length of neurite process in in vitro culture system enough to conclude about actual duration of neurogenesis and effect of EZH1 on it?

RESPONSE: We agree with R1, change in neurite length can also indicate effects on cytoskeletal dynamics, and extra- intra- cellular signaling cues. Although we do not know the molecular mechanism underlying changes in neurite length associated with EZH1 LOF and GOF variants, changes in neurite length combined with the observed effects on neuronal differentiation in the chick embryo neural tube and hPSC derived neuronal and forebrain organoid models, support that EZH1 activity is important for neuronal differentiation. The revised manuscript includes new data supporting that EZH1 LOF and GOF alter neuronal differentiation (*Fig 5c, d, f, g, Supplementary Fig b-e and Supplementary Fig 7*) in a way that fits with a model of delayed and accelerated neurogenesis respectively. However, we acknowledge that more detailed analyses are needed to confirm that the cause of neurogenesis defects by EZH1 variants are due to changes in the timing of neurogenesis and we have revised the manuscript accordingly to avoid overstatements.

4. Lines 322-323: Smaller area is not always equivalent to less postmitotic neurons. The authors need to quantitate the neuronal density or its alterations as an effect of the mutations before coming to this conclusion.

RESPONSE: We agree that smaller area is not always equivalent to less cells (smaller cell size or less extracellular matrix can also lead to smaller area). However, the quantification of the mantle zone area is an accepted estimate of number of cells in the field (see (Le Dreau *et al.*, 2014), especially when the effect is large as in the case of EZH1shRNA, and the cell markers used for quantification are cytoplasmic, like HuC/D. Note that quantifying number of cells based on HuC/D cytoplasmic signal involves challenges defining the limits of individual neurons (this is one of the reasons why we rather quantify the mantle zone area). Nonetheless, following R1's suggestion, we have quantified the number of postmitotic cells in a subset of Scrb and shEZH1 embryos and results confirm that the reduction in MZ area with shEZH1 also corresponds with a decrease in postmitotic neurons (*Figure R2*).

Minor comments:

1. Supplementary Figure 3: EZH1+ cells are not clear. Adding a magnified image showing colocalization will be helpful.

RESPONSE: We have added magnified images to the revised manuscript (*Supplementary Fig 5a*).

Reviewer #2 (Remarks to the Author):

In this paper, Gracia-Diaz and colleagues report on the expression and role of Polycomb Repressive Complex 2 (PRC2) member EZH1 during neural development and its putative importance in human neurodevelopmental disorders. First, they report the identification of mutations in the EZH1 gene in 17 patients exhibiting apparently overlapping neuropsychiatric symptoms. In an attempt to provide further understanding of the molecular pathology caused by these mutations, the authors then use results from ChIP-Seq and Western Blotting experiments to suggest that some EZH1 genetic alterations found in patients are gain-of-function mutations that cause histone hypermethylation. Next, the authors embark on a series of experiments to phenotypically (histologically) analyze several cellular models of neurodevelopment (chick neural tube, differentiating human neurons in 2D culture and cerebral organoids) to suggest that neuronal differentiation is impaired after EZH1 expression levels or sequence are genetically manipulated.

Very little knowledge exists in the literature on the role of EZH1, particularly in the context of neurodevelopment, and evidence for its importance in clinical cases of neurological disorders is lacking. Therefore, this manuscript has the potential to fill important gaps of knowledge. However,

the specific contributions of EZH1 in three fundamental processes that govern neural development (progenitor proliferation, neuronal differentiation and cellular migration) were not parsed out in this paper and therefore the conclusions drawn, especially in regard to the role of EZH1 during neuronal differentiation, are still immature and must be corroborated by further experimentation and controls.

We greatly appreciate the thorough review of our manuscript provided by R2. Their comments have significantly contributed to strengthen our findings on the effects of EZH1 variants in EZH1 molecular and neurobiological function, and to provide accuracy to the description of results and conclusions in our manuscript. Following R2's suggestions, our revised manuscript includes new data supporting that EZH1 LOF variants impair EZH1 expression and EZH1 GOF variants that we experimentally test increase overall H3K27me3 levels at different degrees and through different mechanisms. We also include new data supporting that EZH1 variation affects neurogenesis. Although, these data suggest that defects shown by EZH1 LOF and GOF fit with a model of delayed or accelerated neurogenesis, we acknowledge that further work is needed to dissect the exact mechanism by which EZH1 regulates neurogenesis, thus we have also revised the manuscript to avoid overstatements about effects of EZH1 in "timing" of neurogenesis. With the revised manuscript we hope to disseminate the molecular diagnosis for novel dominant and recessive NDDs and to highlight the relevance of EZH1 in nervous system development and function which has remained underexplored so far.

A summary of the main points that require further experiments or clarification is provided below.

Major comments:

1) Why did the authors not include parents or unaffected siblings of matching sex as controls in the experiments in Fig. 3e-f? The use of unrelated controls is less than ideal, as a large inter-individual variability in gene expression is expected in these experiments.

RESPONSE: We agree with R2 that parents and unaffected siblings are better controls, but unfortunately, we were unable to obtain cells from these individuals. However, from all of the EZH1 WB and qPCR analysis we have made in EZH1^{+/+} control cells (including LCLs, Fibroblasts, hPSCs, NPCs and neurons in *Fig 2d, f, Fig 3d, Supplementary Fig 2a, d, Supplementary Fig 4d, f, g and Supplementary Fig 6f*) we never observe in control cells the low (or undetectable) EZH1 levels of the P12 cells (and KO and now also E485X cells). To illustrate this better we now include in *Fig. 2f and Supplementary Fig. 2d* WB and RT-qPCR analyses of multiple control cell lines showing that EZH1 levels are similar across them and only reduced to near-undetectable levels in P12 patient cell lines. Furthermore, RT-PCR results in *Figure 2e* show that exon 10 containing transcripts are only present in Ctrl cells but not in P12, which reduces possibilities of full length EZH1 expression and further supports that qPCR and WB quantification results are not owing to cell-to-cell variability in EZH1 expression.

2) The reportedly functional data (line 166 onwards) implicating the missense variants with loss-of-function of EZH1 is based on molecular modeling. In the absence of further confirmatory experiments in genetically modified cell lines (or patient-derived cells), these data are just suggestive and probably incomplete. This problem is compounded by the absence of molecular modeling data based on the resolved EZH1 structure itself (the EZH2 structure was used for evaluating most variants in the paper).

RESPONSE: This comment includes concerns related to several aspects of LOF and GOF variants that we address below:

- a) The first half of the line 166-onwards (now line 181-onwards), provides pathogenetic predictive scores and describes experiments indicating that biallelic variants (and not missense variants) result in EZH1 loss-of-function (LOF). To provide further experimental evidence that biallelic variants result in EZH1 LOF, we have generated homozygous EZH1 p.E485X hPSC lines and analyzed *EZH1* mRNA and protein levels. As predicted by the premature stop codon introduced by this nonsense mutation, EZH1 expression is nearly undetectable in cells carrying EZH1 p.E485X (*Fig 2d* and *Supplementary Fig 2a*). Furthermore, while revising the manuscript we have identified an additional patient carrying a homozygous EZH1 nonsense variant (p.Q413X) and neurodevelopmental delay (P19) and a relative of P13-16 patients with the same p.E485X mutation and neurodevelopmental disorder (P17). Although we were unable to obtain cells with these and the p.R258X variant due to various technical and logistic reasons, we consider that, the loss of expression predicted for nonsense variants and experimental evidence we provide for P12 and p.E485X are strong support that biallelic truncating variants in EZH1 have a LOF effect.
- b) The second half of this section (line 208-251) describes EZH1 missense variants and includes molecular modeling predicting that missense variants affect EZH1 structure. As suggested by R2, we now include molecular modeling on the experimental EZH1 structure (Grau *et al.*, 2021) for the variants falling within resolved regions (E438D, K612M and A678G). These models show that structural changes caused by the new variants are similar to those observed when EZH2 based model was used (*Supplementary Fig. 3b*). The variants R728G, Q731E and L735F fall within a gap of the experimental EZH1 structure. Thus, we can only predict their structural effect on the model of EZH1-SET domain we generated using the 3D structure of EZH2-SET (PDB:5hyn(Justin *et al.*, 2016)) as template. The similarity on the aminoacid sequence between EZH1 and EZH2 SET domains (which are 94% similar), and the similarity of the structural prediction for E438D, K612M and A678G regardless of the template used, support the predictions for R728G, Q731E and L735F shown in *Supplementary Fig 3b*.

However, note that the molecular modeling does not provide information about the effect of the variants on EZH1 function (i.e. predicted structural changes do not inform about gain-of-function effect). Indeed, as we mention in the discussion (line 539-549) we do not discard that different missense variants may have different effects on EZH1 function.

- c) We only conclude that missense A678G and Q731E variants have GOF effects from experiments shown in *Fig 3* and described in a different section of the manuscript (now line 273-333). Following R2's suggestion we now include H3K27me3 WB analyses in cells carrying A678G and Q731E in heterozygosity (*Fig 3d* and *Supplementary Fig 4d and e*). Interestingly, the increase in H3K27me3 levels in heterozygous conditions is variable: in A678G cells there is only a non-significant trend to increased H3K27me3 while Q731E cells show a significant 50% increase compared to their isogenic controls (*Fig 3d*). However, these differences are only seen when EZH2 expression levels are reduced (by neuronal differentiation), suggesting that most H3K27me3 is deposited by EZH2 during development, which in turn, interferes with the analyses of overall H3K27me3 levels by WB in cells carrying EZH1 variants. HMT assays in *Fig 3f* and *g* overcome this problem because there is no EZH2 in these experiments. Furthermore, the different effect of A678G and Q731E variants on unmethylated and dimethylated substrates, may explain the more pronounced increase of overall H3K27me3 levels in heterozygous Q731E neurons compared to A678G. In follow up studies that we hope to complete in coming years we are also trying to test these findings in cells, by identify EZH1 specific genomic targets and quantifying their H3K27me3 levels (by ChIP-seq) to precisely dissect the effects of each heterozygous EZH1 variant in H3K27me3.

3) The MRI data in Fig. 1d is misleading because the text (lines 230-238) indicates that most patients do not display abnormal MRI findings. Therefore, these data should be moved to a supplementary figure, and the legend must clearly indicate that these findings are not the norm for most patients.

RESPONSE: We appreciate this suggestion. We have now moved the MRIs to *Supplementary Fig 1c*.

4) The section starting on line 228 attempts to describe the clinical presentation of neurodevelopmental abnormalities in all patients. This analysis makes it clear that the variability among patients is very large and no correlation was found with genotype (and probably could not, based on the small sample sizes for each mutation type). Therefore, I do not see this section as particularly revealing of the pathophysiology and disease mechanisms, which seem to be the paper's focus. I suggest that the section describing clinical characteristics be moved to the supplementary information or methods section. I also disagree with the last sentence in this section ("Additionally, differences in clinical features or severity between patients does (sic.) not correlate with the type or zygosity of EZH1 mutations") as no rigorous testing of correlation was performed (and probably could not, based on the small sample sizes).

RESPONSE: Following R2's suggestion, we have summarized this section and included more careful statements when comparing heterozygous and biallelic patients. However, given that this is the first report of a neurodevelopmental disease associated with EZH1 mutations, we consider clinically relevant to describe major clinical features (especially those that are comparable with EZH2 associated Weaver syndrome) and to highlight that with the current clinical information patients carrying heterozygous missense and biallelic LOF variants are clinically indistinguishable.

(5) The experiments in Fig. 3a-b,d, in which EZH1 protein and histone methylation levels were assayed in a neural cell line over-expressing some missense EZH1 patient variants, are lacking quantification. This is an important piece of information. Proper replication and statistical analyses along with appropriate quantification (via for example band intensity analysis or preferably flow-cytometry) must be included.

RESPONSE: We now include additional replicates, band intensity quantifications and statistical analysis of EZH1 and EZH2 levels (*Fig 3a* and *Supplementary Fig 4a*) and H3K27me3 (*Fig 3 b*).

(6) Likewise, the ChIP-Seq data in Fig. 3c is not replicated and statistical testing is lacking; a purely qualitative assessment is not ideal in this case, as these data are being presented as supporting evidence that the missense mutation increases EZH1's histone methylation activity.

RESPONSE: The enrichment plot and heatmap in *Fig 3c* shows intensities of sequencing reads combined from 3 independent replicates (ChIPseq performed in 3 independently transduced cell cultures). In addition, following R2's recommendation, we now include a statistical analysis that indicates the increase in H3K27me3 signal in A678G is statistically significant (*Fig 3c* legend and methods). Furthermore, in *Supplementary Fig 4b* we show the enrichment plots and heatmaps of each of the 3 independent replicates, which support reproducibility of increased H3K27me3 signal when EZH1-A678G is overexpressed.

(7) Although the experiments in Fig. 3e-f were nicely conducted, the quantification in Fig. 3f

indicates the use of two replicates (n=2) and statistical power analysis is likely to indicate lower-than-expected sampling to achieve statistical significance in ANOVA followed by Dunn's post-hoc test.

RESPONSE: We appreciate this observation and apologize for the error in the figure legend stating that data in *Fig 3f-g* was analyzed with one-way ANOVA instead of two-way ANOVA that we used. Two-way ANOVA with Dunnett's post hoc analysis test for multiple comparison indicates that the effect of A678G and Q731E on methylation is significantly different to the effect of EZH1wt, even with the 2 replicates included in *Fig 3g*. As shown in other publications (Margueron *et al.*, 2008; Lee *et al.*, 2018a; Lee *et al.*, 2018b; Lee *et al.*, 2019), this type of experiments have little variability and are usually done in duplication with increasing PRC2 concentrations each as in *Fig 3g*.

(8) The design and interpretation of Fig. 4 results have many problems:
- the authors should improve their description of the developmental model used (chick neural tube), including a better description that HuC/D is used as a marker for the mantle zone (MZ) and SOX2/9 are used as markers of ventricular zone (VZ) neural progenitor cells.

RESPONSE: We thank the reviewer for this suggestion and have revised the manuscript accordingly to include a better description of the model, the electroporation method and the markers used to label MZ neurons and VZ progenitors.

- Suppl. Fig. 3 very clearly shows that EZH1 is expressed predominantly in the MZ. This makes the experiments in which the expression of EZH1 is manipulated in the chick neural tube's VZ very hard to interpret. The authors should comment on this matter and, if the results are kept in the paper, acknowledge this caveat.

RESPONSE: We have revised the manuscript to better explain that the electroporation of the EZH1-shRNA, which occurs in neural progenitor cells only, may prevent the upregulation of EZH1 in electroporated neural progenitors undergoing neuronal differentiation associated with neuronal differentiation (line 362-365).

- why was SOX9 used as a VZ marker for the shRNA experiments while SOX2 was used as a VZ marker for the EZH1 overexpression experiments?

RESPONSE: The only reason of using SOX2 or SOX9 was the availability of each antibody in the lab at the time the experiments were conducted. Both, SOX2 and SOX9 label neural progenitors in the ventricular zone.

- how can one differentiate between GFP+ electroporated VZ cells that then migrate to and differentiate in the MZ from the scenario in which GFP+ MZ cells are being electroporated in the first place?

RESPONSE: At the developmental stage that neural tubes were electroporated (HH14 ~54hpf), neural progenitors are the only cells in the neural tube (see absence of neurons, labeled by anti-HuC/D, at HH14 in *Figure R1*), therefore electroporation of MZ neurons is not a possible scenario. We acknowledge that this detail was not well described in the submitted manuscript, which we have now revised accordingly.

- the data in Fig. 4c (fewer GFP+ cells in MZ and fewer HuC/D cells in MZ in shEZH1 neural tubes as compared to scrambled controls) is presented as evidence that "EZH1 regulates the balance

between neural progenitor cells proliferation and differentiation during neural development” (line 329). This is certainly not the only explanation. EZH1 may impact VZ progenitor proliferation and/or VZ-to-MZ migration and/or progenitor-to-neuron differentiation in the MZ. This model and experiments do not allow us to parse out between these three possibilities. Similarly, the EZH1 overexpression experiments are being presented as evidence that “EZH1 induces the *differentiation* of neural progenitor cells” (line 328), which is certainly unwarranted by the data.

RESPONSE: We appreciate R2’s alternative interpretations of results in *Fig 4*, which result, in part, from the poor description of the chick embryo neural tube electroporation method we provided in the original submission. The revised manuscript addresses this issue. In particular, it is important to note that only neural progenitors can be electroporated at HH14 because there are no neurons in HH14 neural tubes (see *Figure R1* and (Saade *et al.*, 2013; Le Dreau *et al.*, 2014; Saade *et al.*, 2020)). Thus, at the analysis time point, which is 48h after the electroporation, we see the progeny of the electroporated neural progenitors labeled with EGFP. In control conditions (Scrb or Ctrl) EGFP cells are evenly distributed between the SOX2+ neural progenitor cell pool in the VZ and the HuC/D+ neurons in the MZ, as expected by the switch from proliferative to neurogenic divisions of neural progenitor cells that occurs at ~HH18 (Saade *et al.*, 2013), (~16h after electroporation). A deviation from this even distribution, like decreased and increased proportion of MZ versus VZ localized EGFP+ cells in shEZH1 and EZH1 electroporated neural tubes respectively, indicates effects in neuronal differentiation. Furthermore, in the case of shEZH1 and EZH1 electroporation, neural progenitors (SOX2/9+) and neurons (HuC/D+) are correctly localized in the VZ and MZ respectively, thus indicating that they affect events that lead to neuronal differentiation in a coordinated way. If shEZH1 or EZH1 electroporations were affecting proliferation of neural progenitors or their delamination and migration to the MZ independently, we would expect ectopic localization of HuC/D+ neurons in the VZ and/or SOX2/9+ neural progenitors in the MZ (see (Akizu *et al.*, 2016; Wilmerding *et al.*, 2021) for examples). Nonetheless, we agree that concluding that ‘EZH1 regulates the balance between neural progenitors and neurons’ is imprecise, and have revised this, and similar sentences, to avoid overinterpreting conclusions. Furthermore, we have performed additional experiments that show similar amounts of mitotic events and low apoptotic cell numbers for neural tubes analyzed 48h after the electroporation with controls, shEZH1 or EZH1 (*Supplementary Fig 5b-e*). Although more detailed analysis including a time course could further clarify the exact mechanism by which EZH1 affects neuronal differentiation, this is out of the scope of this manuscript and also challenging due to the transient and mosaic nature of the model (i.e. earlier analysis windows may not allow shEZH1 interference and electroporated plasmids are diluted at later analysis windows).

- the statistical representation of data in Figs. 4c-d conflates technical replicates (histological sections from the same embryo) with biological replicates, which is inappropriate. Only the biological replicates should be presented and used for statistical purposes (all sections from the same individual should be aggregated into a mean per individual). Based on the SDs, it is likely that none of the mean difference comparisons shown will turn out to be statistically significant after this fix is applied (particularly in Fig. 4d).

RESPONSE: We agree with R2 that only biological replicates (individual embryos) should be used for statistical purposes and the original *Figure 4d* data points do correspond to one embryo each (averaged from the analysis of 2-5 sections per embryo). However, we inadvertently plotted all the quantified sections in the *Figure 4c* graphs, which we have now corrected to represent individual embryos as replicates. We have also corrected the figure legend accordingly.

- in the absence of molecular markers, it is not possible to define the exact location and cell types expressing EZH1 in the developing chick neural tube (Fig. 4a).

RESPONSE: We agree that we need colocalization with molecular markers to define the exact cell types that express EZH1. However, the spatial segregation of neural progenitors and neurons, located in the VZ and MZ of the neural tube respectively (See *Figure R1*), allow us to conclude that EZH1 expression is enriched in the MZ, which is constituted by postmitotic neurons at the analyzed time points.

Nonetheless, we have tried to follow R2's suggestion to complement data in Fig 4a with neuronal marker colocalization studies, but unfortunately, we no longer have the EZH1 antibody (from Reinberg lab) we used for Fig 4a and immunofluorescent stainings with 3 new EZH1 antibodies we obtained have all failed (they either show background signal or no signal). However, to further support that EZH1 expression increases as neurogenesis progresses in the chick embryo neural tube, we now include in Fig 4a additional EZH1 immunofluorescence stainings we performed in the past with the original antibody. As shown in the revised Fig 4a, EZH1 is undetectable in HH12 (~49hpf) embryo neural tubes, which are only comprised by neural progenitors. HH23 (~96hpf) embryo neural tubes show EZH1 staining in their thin MZ; and in HH30 (~156hpf) neural tubes, EZH1 is also expressed in the MZ which is composed of more neurons and thus larger than the HH23 MZ. Please, note the similarity between EZH1 expression pattern in Fig 4a and HuC/D expression pattern in Fig R1. We hope that these additional data and the revised description of the chick embryo neural tube development model in the manuscript addresses R2's concern.

(9) In the experiments conducted in human neurons differentiated from hPSCs (Fig. 5), the CRISPR-derived mutant lines contain clinically relevant mutations only for the heterozygous variant. Why wasn't the loss-of-function(-/-) line produced with a clinically relevant truncating variant?

RESPONSE: technical and logistic reasons precluded us to include clinically relevant mutations for LOF in this manuscript. Precisely, the first two patients we identified were two patients with EZH1 missense heterozygous variants (A678G and L735F). The next was P12, with compound heterozygous splice and large deletion (exon 8-12) variants (confirmed to deplete EZH1 expression later). Given that we were unable to precisely reproduce this genotype using CRISPR in hPSCs, we anticipated that a generic EZH1 knock out hPSC line would reproduce the functional consequence of LOF variants in P12 and started working with EZH1^{-/-} we generated by CRISPR (and the EZH1^{+A678G} hPSCs that we also generated). Right after, the COVID19 pandemic hit the world and our efforts to collect patient cells for reprogramming, and to identify new patients with EZH1 mutations were halted. Luckily, we were able keep going with our functional studies with the EZH1^{-/-} and EZH1^{+A678G} and determine that EZH1 is important for neurogenesis and variants are the likely cause NDD in identified patients. Since 2021, we have exponentially increased the identification of new patients (including 2 that we have identified during the revision of our manuscript) and we are making efforts to collect samples from them and generate new iPSC models with additional mutations. The two that we just generated (P12 iPSC, E485X) are included in the revised version of the manuscript for molecular validation, but analyzing neurodevelopmental phenotypes with proper controls will take some time, and we do not think the conclusions will be as different as to justify holding our findings from dissemination.

(10) The EdU+ cell counts in Fig. 5d are problematic, because the experiment is not controlled for the vagaries of the percentages of NPCs versus neurons generated in these types of 2D culture. Therefore, a measure of the diversity of cell types in these cultures must be included and

factored into the calculations. Also, cytometry-based methods for determining the percentages of EdU+ cells are more appropriate.

RESPONSE: We completely agree with R2. Indeed, the heterogeneous nature of neuronal differentiations in 2D cultures was the major factor that motivated our forebrain organoid experiments, which results are consistent with the defects in neuronal differentiation we found in 2D cultures (and chick embryo neural tube experiments). Furthermore, we consider that addressing the heterogeneity issue by using brain organoids that allow quantifying the number of neurons over the neural progenitors in the adjacent VZ for normalization across rosettes is more accurate than factoring 2D cultures for cell type specific markers analyzed in a parallel culture plate (which generates other problems). However, as R2 suggests, we do complement data in Fig 5 with a flow cytometry analysis that we developed to combine a proliferation marker (Ki67), a neural progenitor marker (SOX2) and a postmitotic neuron marker (HuC/D) in the same analysis. In our test experiment with wild type H9 ESCs, we confirmed that the major population of cells at day 0 of differentiation is Ki67+SOX2+ and HuC/D-, as expected for neural progenitors (Fig R3). As differentiation goes on (at day 2 and 5) there is a progressive increase of HuC/D+ and Ki67-SOX2- differentiating neurons (and another uncharacterized population that is triple negative) (Fig R3). We then run this analysis at D0 and D5 in EZH^{+/+}, EZH^{-/-} and EZH^{+/A678G} cells and confirmed that EZH1^{-/-} fail to generate as many HuC/D+ Ki67-SOX2- differentiating neurons. In addition to being more quantitative than our immunofluorescent quantifications, **this flow cytometry analysis addresses the heterogeneity issue of the 2D cultures, by limiting the analysis to SOX2+ progenitors and HuC/D+ neurons.**

Figure R3: Flow cytometry analysis of proliferating and differentiating hPSC-derived neural populations at day 0, 2 and 5 after induction to differentiate. Neural progenitors are Ki67+HuC/D- (left) and Ki67+SOX2+ (right) and differentiating neurons are HuCD+/Ki67- (left). Contour plots show a progressive increase of differentiated neuron population and a decrease of neural progenitors from D0 to D5.

(11) Although I consider the data on neurite length in Fig. 5d quite interesting and revealing of the potential cellular mechanisms derived from EZH1 loss-of-function, I disagree with the interpretation that these data reveal defects in neuronal *differentiation*. Neurite extension and growth is but one of many mechanisms during cellular differentiation and using that as a proxy for neuronal maturation is a dated approach. Rather, the authors should evaluate the expression of known neuronal maturation markers or analyze the transcriptomic landscape of these cells during differentiation induction in vitro (via RNA-Seq followed by RNA velocity analysis or pseudotime differentiation analysis). Moreover, it is hard to define whether differentiation is impaired if the cellular composition of differentiating cultures is not considered during the interpretation of these results, as pointed out above.

RESPONSE: We also agree. Change in neurite length can also be the result of a differentiation independent process, and an interesting phenotype to follow up. However, combined with the original and revised experiments in Fig 5 (and Fig 4 and 6), changes in neurite length is consistent with altered neuronal differentiation phenotypes we observe with EZH1 LOF and GOF. Following R2's suggestion, to further support that EZH1 variants alter neuronal differentiation, we performed

a bulk RNAseq experiment in 8-week-old neuronal cultures. Data show an enrichment of neural stem cell gene sets among $EZH1^{-/-}$ upregulated genes and late-born neuron gene sets in $EZH1^{+/A678G}$ (Fig 5g), which provides additional evidence supporting that EZH1 is important for neuronal differentiation and altered by LOF and GOF variants.

(12) Fig. 6a attempts at measuring the thickness of SOX2+ VZ-like neuroepithelial structures in cerebral organoids. However, these organoid derivation protocols lead to organoids with many such structures, which often differ in size and width (as can be seen in Fig. 6b). Moreover, the width of a particular VZ-like structure cannot be easily evaluated unless the experimenter analyses serial histological sections of each organoid. How did the authors select and quantify the VZ structure widths?

RESPONSE: We agree with R2 that there are significant challenges for quantitative analysis of brain organoids due to limitation of current technologies. To account for the variability, for each genotype, we sectioned at least three organoids for analysis. For each organoid, we immunostained and imaged at least three sections distributed evenly from top to bottom of each organoid. During confocal imaging, only rosettes at the edge of the organoids were captured to avoid the variability caused by the limited nutrient access in the center. Extremely small or big rosettes were avoided. Rosettes without clear lumen were also avoided as they may locate in very top/bottom region of the organoid. The distance from the apical to the basal lamina in a randomly selected position of each rosette in the section was measured for at least 3 sections per organoid. We have revised the manuscript to provide the detail quantification methodology (line 928-936).

(13) I disagree with the conclusion from Fig. 6a that the larger width of VZ structures in organoids with EZH1 loss-of-function(-/-) reveal prolonged proliferation rates and deficits in neuronal differentiation (line 386). Either of these processes may be altered, but these experiments do not discriminate between both possibilities.

RESPONSE: We agree with reviewer with the two possibilities and include them in the revised manuscript (line 493-494).

(14) The results in Fig. 6b show a deficit in CTIP2+ cortical-type neurons in $EZH1^{-/-}$ organoids and an apparent increase in SATB2+ neurons in $EZH1$ “gain-of-function” organoids. However, the SATB2 staining seems odd. The percentages of SATB2+ neurons in the right-most panels are very high, which is unusual for 60-day organoids derived as per the protocol used. Moreover, immunostaining seems to label every cell between VZ-like neuroepithelia in “batch 2”. It was impossible to judge whether SATB2+ staining was overlapping with CTIP2+ staining in these cells, but it appears to be so. Therefore, proper controls, higher magnification panels, better quantification, and more replicates must be included to support the labeling quantifications in the right graphs.

RESPONSE: At Day 60, we indeed observed high amount of SATB2+ cells only in $EZH1^{+/A678G}$ mutant organoids, but not in $EZH1^{+/+}$ or $EZH1^{-/-}$ organoids as expected for organoids derived with this protocol (and noted by R2). These results were consistent in the two batches of organoids we generated and the difference between $EZH1^{+/+}$ and $EZH1^{+/A678G}$ in the 10-13 organoids we quantified was statistically significant, thus indicating that the unusual high numbers of SATB2+ cells is a relevant phenotype specific to $EZH1^{+/A678G}$ organoids. Also note that in order to account for variability we normalize the amount of SATB2+ neurons (or CTIP2+) by the amount of SOX2+ neural progenitors in the underlying VZ. In addition, our previously published protocol (Qian *et al.*, 2016) used feeder-iPSCs, rather than feeder-free hESCs and different dual inhibitors in the

current study. These factors could slightly change the timeline of developmental trajectory, meaning the appearance of certain types of neurons may be earlier or later than previously published and highlighting the relevance of comparing results with internal controls as we do, rather than with published examples. We appreciate R2's suggestion to include higher magnification panels in *Fig 6b*, which we do alongside separated channels for CTIP2 and SATB2 for better visualization (*Fig 6b*).

Also, the protocol for organoid derivation (line 629) appears to conflict with the data reported in Fig. 6. If the differentiation factors (GDNF, BDNF, etc.) are added to the medium at day 70 in vitro, how do organoids at 60 days in vitro (Fig. 6) exhibit such high levels of differentiated cortical neurons?

RESPONSE: Once patterned, the cortical organoids follow the endogenous neurogenesis trajectory generating cortical neurons without adding any exogenous differentiation factors. GDNF, BDNF, cAMP ect. are factors that promote neuronal maturation, not neurogenesis. Regardless, for this study we did not grow organoids passed day 60, therefore we have eliminated the day 70 media change in the revised method section, we apologize for creating confusion.

More importantly, the conclusion drawn from these data -- that the different numbers of cortical neurons in genetically manipulated organoids indicate that EZH1 variants alter the timing of neurogenesis – is unwarranted. Many processes may lead to reduced/augmented numbers of cortical neurons, including abnormal NPC proliferation, migration, or cell death.

RESPONSE: The reviewer raised several possibilities to explain the phenotypes we observed. Among these possibilities, we tested cell death with cleaved Casp 3 and results showed no differences between genotypes, at least at Day 60 (*Supplementary Fig e*). Furthermore, we discard gross defects in migration (as isolated from the overall neurogenesis trajectory), because in such case, we would expect ectopically localized SOX2+ neural progenitors (in MZ) or CTIP2+ and SATB2+ neurons (in VZ) in mutant organoids, which we do not. Note, that with current forebrain organoid protocols it is difficult to assess neuronal layer specific migration defects, given that most protocols result in organoids without well-defined cortical layers. Thus, we can not discard layer specific migration defects as a possibility with currently available tools. Regardless, our data, which is replicated in n=10-13 organoids derived from two different batches, uses internal controls and normalized quantifications to account for intra and inter-organoid variability, indicates deficits in neurogenesis of CTIP2 and SATB2 expressing cortical layer neurons associated with EZH1 LOF and GOF. We acknowledge that more thorough analyses, some of which are not possible with current technologies, are required to confirm that “timing” is the factor that affects neurogenesis in EZH1^{-/-} and EZH1^{+/^{A678G}} organoids, and therefore we have revised our manuscript (including the title) to avoid overinterpretations accordingly. Importantly, the consistent defects in neuronal differentiation we observe in the three model systems analyzed (chick neural tube, 2D neural cultures and forebrain organoids) support that EZH1 LOF and GOF impair neurogenesis, and thus mutations in EZH1 are the likely cause of novel dominant and recessive neurodevelopmental disorders.

Minor comments:

(1) Why did the authors perform RT-PCR and WB analysis on EZH1 for P12 only (lines 184-186 and in Fig. 2e-f)? Why were the other biallelic truncating variant individuals in the cohort not included?

RESPONSE: As indicated above (response to major concern 9), due to technical and logistic reasons we were unable to obtain cells from other patients with biallelic truncating variants or to generate them with CRISPR until recently.

(2) The expression levels in control in Fig. 2e should have a normalized *MEAN* of 1, but each individual replicate measurement must be indicated, rather than normalized to 1.

RESPONSE: We have corrected this (now included in *Supplementary Fig 2d*)

(3) Sequencing data and data supporting the genetic mosaicism in the father of one of the heterozygous affected individuals should be presented (related to lines 190-194) rather than just described in the text.

RESPONSE: We have included Sanger chromatograms of the affected child and parents, in *Supplementary Fig 2e*.

(4) Some wording seems short of scientific precision, such as “diagnostic odyssey” and missense tolerancy (Fig. 1). Also, “catalyzes most of the H3K27me3” should be replaced with “catalyzes most of the methylation at H3K27” (there are similar instances throughout the rest of the manuscript).

RESPONSE: We appreciate R2’s text editing suggestions and corrected them accordingly. For the suggestion on “catalyzing methylation at H3K27me3 (vs catalyzing H3K27me3)”, we can find examples of both in the scientific literature “catalyzing methylation of H3K27” (Lee *et al.*, 2006; Pasini *et al.*, 2007; Kim *et al.*, 2018) and “catalyzing H3K27me3” (Lee *et al.*, 2018a) (Escobar *et al.*, 2019). Thus, to keep consistency and improve reading flow, we prefer to maintain H3K27me3 across the manuscript (rather than alternating between H3K27me3 and methylation of H3K27 depending on the context). We also substitute “missense tolerancy”, for “missense tolerance”, which is a score calculated from data in gnomAD and provided in MetaDome (Wiel *et al.*, 2019).

Reviewer #3 (Remarks to the Author):

Gracia-Diaz and Zhou et al. reported that mutations in EZH1, which is part of PRC2 subunits, were identified from 17 individuals with neurodevelopmental delay. They found that biallelic recessive EZH1 variants led to LOF of its methylation activity, whereas heterozygous dominant variants lead to GOF. Interestingly, using a chick embryo neural tube model, they found that EZH1 down and upregulation both impaired neuron differentiation, but in opposite directions. They further supported this claim using hPSC derived isogenic 2D NPCs by showing EZH1 LOF variants led to delay neuron differentiation, whereas EZH1 GOF variant such as EZH1+/A678G promoted neuronal maturation. Lastly, they used a 3D forebrain organoid model to demonstrate that EZH1 GOF (EZH1+/A678G) and LOF variants affect neuron differentiation by specifically impacting upper layer neuron and deep layer neuron production, respectively.

We thank reviewer 3 for recognizing the novelty of our findings and raising issues that needed clarification or correction to improve the quality of the manuscript.

This paper reported novel findings on the role of EZH1 on neurodevelopmental delay. We have several minor comments for the authors to address.

Minor comments:

1. Typo in line 968, should be hPSC instead of hPCS.

RESPONSE: Thank you for catching this, we have now corrected the typo.

2. For Fig. 2f, EZH2 level from P12(1) but not P12(2) is reduced compared to control. Can you explain this inconsistency?

RESPONSE: EZH2 expression levels vary with the proliferative rate of cells (Varambally *et al.*, 2002; Bracken *et al.*, 2003; Wassef *et al.*, 2019), and thus a variation in the proportion of proliferating cells can contribute to EZH2 expression level variability between cell cultures, especially if they are not synchronized for the cell cycle. Furthermore, in the case of hPSC cultures, spontaneous differentiation events, that are variable between wells and cell lines/clones, is an important factor to consider when analyzing the expression level of cell cycle associated genes, like EZH2. After quantifying the expression of EZH2 in several replicates of control and P12 cultures by qPCR and WB, we show that, despite the variability between P12(1) and P12(2), there are no significant changes in the levels of EZH2 expression between control and mutant cells (*Fig 2f* and *Supplementary Fig 2d*).

3. Do you have references to support the claim in line 167?

RESPONSE: Missense Z-score and LOF o/e rate are extracted from GnomAD v2.1.1 (Karczewski *et al.*, 2020). We now include references in lines 181-82.

4. Please add statistics for Fig.3e L735 vs WT comparison to support claim in line 296.

RESPONSE: We now add statistics for all pairwise comparisons and revise the manuscript accordingly.

5. For Fig.5c and Fig.6, for the control line, did you use H9 unedited or H9+/+ synonymous lines as was indicated in supplemental Fig. 4e, which would then represent true isogenic with EZH1+/A678G, for 2D NPC and 3D forebrain organoid generation?

RESPONSE: Data in *Fig 5* is generated with both, H9 unedited and H9+/+ synonymous, and shown combined under the EZH1^{+/+} label. With the goal to simplify the experimental load in organoid generation and given that we did not observe significant phenotype differences between H9 unedited and H9+/+ synonymous in monolayer experiments, data in *Fig 6* is from H9 unedited.

6. Based on supplemental table 1, the p.A678G variant was identified in the male patient P4. Why did you use H9, which is a female hESC line, to model this variant in Fig.5 and Fig.6?

RESPONSE: The only reason we used H9 cells to model the disease is that at the conception of the project we had the most experience CRISPR editing and differentiating H9 hESCs. Although our cohort size is too small to assess if gender influences phenotypes of the patients, there are males and females within the two cohorts (heterozygous and biallelic) suggesting that EZH1 variants affect EZH1 function indistinctive of the gender. Nevertheless, we understand the relevance of considering gender as a variable (especially when H3K27me3 is involved in X chromosome inactivation) and thus, we are currently generating male hPSC lines carrying various EZH1 variants to include both, males and females, in future studies.

REFERENCES

- Akizu N, Garcia MA, Estaras C, Fueyo R, Badosa C, de la Cruz X, *et al.* EZH2 regulates neuroepithelium structure and neuroblast proliferation by repressing p21. *Open Biol* 2016; 6(4): 150227.
- Bracken AP, Pasini D, Capra M, Prosperini E, Colli E, Helin K. EZH2 is downstream of the pRB-E2F pathway, essential for proliferation and amplified in cancer. *EMBO J* 2003; 22(20): 5323-35.
- Escobar TM, Oksuz O, Saldana-Meyer R, Descostes N, Bonasio R, Reinberg D. Active and Repressed Chromatin Domains Exhibit Distinct Nucleosome Segregation during DNA Replication. *Cell* 2019; 179(4): 953-63 e11.
- Grau D, Zhang Y, Lee CH, Valencia-Sanchez M, Zhang J, Wang M, *et al.* Structures of monomeric and dimeric PRC2:EZH1 reveal flexible modules involved in chromatin compaction. *Nat Commun* 2021; 12(1): 714.
- Justin N, Zhang Y, Tarricone C, Martin SR, Chen S, Underwood E, *et al.* Structural basis of oncogenic histone H3K27M inhibition of human polycomb repressive complex 2. *Nat Commun* 2016; 7: 11316.
- Karczewski KJ, Francioli LC, Tiao G, Cummings BB, Alföldi J, Wang Q, *et al.* The mutational constraint spectrum quantified from variation in 141,456 humans. *Nature* 2020; 581(7809): 434-43.
- Kim J, Lee Y, Lu X, Song B, Fong KW, Cao Q, *et al.* Polycomb- and Methylation-Independent Roles of EZH2 as a Transcription Activator. *Cell Rep* 2018; 25(10): 2808-20 e4.
- Le Dreau G, Saade M, Gutierrez-Vallejo I, Marti E. The strength of SMAD1/5 activity determines the mode of stem cell division in the developing spinal cord. *J Cell Biol* 2014; 204(4): 591-605.
- Lee CH, Holder M, Grau D, Saldana-Meyer R, Yu JR, Ganai RA, *et al.* Distinct Stimulatory Mechanisms Regulate the Catalytic Activity of Polycomb Repressive Complex 2. *Mol Cell* 2018a; 70(3): 435-48 e5.
- Lee CH, Yu JR, Granat J, Saldana-Meyer R, Andrade J, LeRoy G, *et al.* Automethylation of PRC2 promotes H3K27 methylation and is impaired in H3K27M pediatric glioma. *Genes Dev* 2019; 33(19-20): 1428-40.
- Lee CH, Yu JR, Kumar S, Jin Y, LeRoy G, Bhanu N, *et al.* Allosteric Activation Dictates PRC2 Activity Independent of Its Recruitment to Chromatin. *Mol Cell* 2018b; 70(3): 422-34 e6.
- Lee TI, Jenner RG, Boyer LA, Guenther MG, Levine SS, Kumar RM, *et al.* Control of developmental regulators by Polycomb in human embryonic stem cells. *Cell* 2006; 125(2): 301-13.
- Margueron R, Li G, Sarma K, Blais A, Zavadil J, Woodcock CL, *et al.* Ezh1 and Ezh2 maintain repressive chromatin through different mechanisms. *Mol Cell* 2008; 32(4): 503-18.
- Pasini D, Bracken AP, Hansen JB, Capillo M, Helin K. The polycomb group protein Suz12 is required for embryonic stem cell differentiation. *Mol Cell Biol* 2007; 27(10): 3769-79.
- Pereira JD, Sansom SN, Smith J, Dobenecker MW, Tarakhovskiy A, Livesey FJ. Ezh2, the histone methyltransferase of PRC2, regulates the balance between self-renewal and differentiation in the cerebral cortex. *Proc Natl Acad Sci U S A* 2010; 107(36): 15957-62.
- Qian X, Nguyen HN, Song MM, Hadiono C, Ogden SC, Hammack C, *et al.* Brain-Region-Specific Organoids Using Mini-bioreactors for Modeling ZIKV Exposure. *Cell* 2016; 165(5): 1238-54.
- Saade M, Ferrero DS, Blanco-Ameijeiras J, Gonzalez-Gobartt E, Flores-Mendez M, Ruiz-Arroyo VM, *et al.* Multimerization of Zika Virus-NS5 Causes Ciliopathy and Forces Premature Neurogenesis. *Cell Stem Cell* 2020; 27(6): 920-36 e8.
- Saade M, Gutierrez-Vallejo I, Le Dreau G, Rabadan MA, Miguez DG, Buceta J, *et al.* Sonic hedgehog signaling switches the mode of division in the developing nervous system. *Cell Rep* 2013; 4(3): 492-503.
- Varambally S, Dhanasekaran SM, Zhou M, Barrette TR, Kumar-Sinha C, Sanda MG, *et al.* The polycomb group protein EZH2 is involved in progression of prostate cancer. *Nature* 2002; 419(6907): 624-9.

Wassef M, Luscan A, Aflaki S, Zielinski D, Jansen P, Baymaz HI, *et al.* EZH1/2 function mostly within canonical PRC2 and exhibit proliferation-dependent redundancy that shapes mutational signatures in cancer. *Proc Natl Acad Sci U S A* 2019; 116(13): 6075-80.

Wiel L, Baakman C, Gilissen D, Veltman JA, Vriend G, Gilissen C. MetaDome: Pathogenicity analysis of genetic variants through aggregation of homologous human protein domains. *Hum Mutat* 2019; 40(8): 1030-8.

Wilmerding A, Rinaldi L, Caruso N, Lo Re L, Bonzom E, Saurin AJ, *et al.* HoxB genes regulate neuronal delamination in the trunk neural tube by controlling the expression of *Lzts1*. *Development* 2021; 148(4).

REVIEWERS' COMMENTS

Reviewer #1 (Remarks to the Author):

The authors have modified the manuscript in both data explanation and analyses, addressing majority of the reviewers' comments as feasible. I do not have any further comments to add.

Reviewer #2 (Remarks to the Author):

The authors have very carefully analyzed my comments and performed a thorough review of the manuscript, with the addition of new data and clarifications in the text that fully resolve the issues I indicated in my original comments.

Reviewer #3 (Remarks to the Author):

The authors have responded to our criticisms and those of the others reviewers, in my opinion.